

# Ice Crystal Characterization in Cirrus Clouds III: Retrieval of Ice Crystal Shape and Roughness from Observations of Halo Displays

Linda Forster[1] and Bernhard Mayer[1,2]

[1]Meteorologisches Institut, Ludwig-Maximilians-Universität, München, Germany.
[2]also at Institut für Physik der Atmosphäre, Deutsches Zentrum für Luft- und Raumfahrt, Oberpfaffenhofen, Germany.

**Correspondence:** Linda Forster (Linda.Forster@physik.lmu.de)

**Abstract.** In this study, which is the third part of the HaloCam series after Forster et al. (2017) and Forster et al. (2020), we present a novel technique to retrieve quantitative information about ice crystal optical and microphysical properties making use of ground-based imaging observations of halo displays. Comparing HaloCam's calibrated RGB images of 22° and 46° halo observations against a look-up table of simulated radiances, this technique allows the retrieval of size and shape of randomly
oriented crystals as well as the fraction of smooth and rough ice crystals for cirrus clouds. We analyzed 4400 HaloCam images between September 2015 and November 2016 showing a visible 22° halo. The optical properties of hexagonal 8-element columns with a mean ice crystal effective radius of about 20 $\mu$m and a mixture of 37% smooth and 63% rough crystals on average best match the HaloCam observations. Implemented on different sites, HaloCam in combination with the machine-learning based halo detection algorithm HaloForest can provide a consistent dataset for climatological studies of ice crystal
properties representing typical cirrus clouds. Representative ice crystal optical properties are required for remote sensing of cirrus clouds as well as climate modeling. Since ground-based passive imaging observations provide information about the forward scattering part of the ice crystal optical properties, the results of this work ideally complement the results of satellite-based and airborne studies.

## 1 Introduction

Cirrus clouds cover about one third of the globe on average (Wylie and Menzel, 1999; Stubenrauch et al., 2006) and consist of small ice crystals. Crystal size, shape, and surface roughness predominantly govern the single scattering properties and thus the radiative forcing of cirrus clouds (e.g. Liou, 1986; Wielicki et al., 1995; Wendisch et al., 2007; Yi et al., 2013). Depending on these microphysical properties, ice clouds may have a net warming or cooling radiative effect for a given ice water content (Stephens et al., 1990). Furthermore, wrong assumptions regarding the ice crystal shape can result in significant errors

in retrievals of optical thickness and cloud microphysical properties using satellite-based shortwave infrared measurements (Mishchenko et al., 1996; Baran et al., 1999; Yang et al., 2015; Holz et al., 2016). The uncertainty in the retrieved cirrus optical thickness and ice crystal effective radius was estimated to more than 50% and 20%, respectively, by Key et al. (2002); Eichler et al. (2009) and Zinner et al. (2016). Ice crystal size and shape also have a significant impact on cloud evolution and





the hydrological cycle (e.g. Jensen et al., 2018). A better understanding of ice crystal microphysical properties and finding representative optical properties is therefore essential to improve remote sensing retrievals of cirrus cloud properties, which in turn helps improve estimates of the radiative forcing of cirrus clouds (e.g. Yang et al., 2015; Liou and Yang, 2016).

Over the past decades the natural distribution of ice crystal shape has been investigated by laboratory studies (Magono
and Lee, 1966; Bailey and Hallett, 2004, 2009) and in situ measurements (Weickmann, 1947; Heymsfield and Platt, 1984; Field et al., 2005; Heymsfield et al., 2013; Magee et al., 2021). Although these methods have been providing more and more detailed information about ice crystal size and shape under various nucleation and growth conditions, they suffer from certain limitations. The nucleation technique used in laboratory studies, for example, can influence the shape of the growing ice crystals and lead to biased results (e.g. Bailey and Hallett, 2012). In situ observations by aircraft probes are spatially limited.
Furthermore, due to the high speed of the aircraft, shattering of larger complex ice crystals at the inlets of the in situ probes is an issue which might cause an artificially increased fraction of small particles (Baran (2012) and references therein).

Therefore, satellite-based methods have been investigated in recent years to retrieve information about ice crystal shape with large spatial and temporal coverage. Retrievals of ice crystal habit from multi-angle satellite observations were pioneered by Baran et al. (1998, 1999) using radiance measurements at two different viewing angles from the Along Track Scanning
Radiometer (ATSR-2). McFarlane and Marchand (2008) present a retrieval using measurements from MISR (Multi-angle Imaging Spectroradiometer) and MODIS (Moderate resolution imaging system) reflectances based on optical properties of single ice crystal habits. Multi-angular polarized reflectances from the Polarization and Directionality of Earth Reflectance (POLDER) have been used to infer information about ice crystal shape (e.g. Descloitres et al., 1998; Chepfer et al., 2001; Baran and Labonnote, 2006; Sun et al., 2006). These studies mainly focus on optically thick cirrus. More recently, polarized
reflectance observations of the airborne Research Scanning Polarimeter (RSP) have been used to retrieve ice crystal aspect ratio and distortion levels for tops of optically thick ice clouds: van Diedenhoven et al. (2012, 2020); van Diedenhoven (2021) found that crystal distortion and aspect ratio increase with cloud top height, leading to decreasing asymmetry parameters.

Investigation of ice crystal shapes in thin cirrus clouds using space- or airborne passive remote sensing is more challenging due to the unknown surface reflectance, especially over land. Wang et al. (2014) and Holz et al. (2016) used a combination of
active and passive remote sensing instruments with co-located MODIS and CALIOP (Cloud Aerosol Lidar with Orthogonal Polarization) observations. Saito et al. (2017) developed an optimal estimation-based algorithm to retrieve optical thickness, effective radius, fraction of (horizontally oriented) plates, and the degree of surface roughness for optically thin ice clouds using CALIOP and IIR (Infrared Imaging Radiometer). The majority of studies implies that ice crystals with roughened surface represent the observations better than crystals with smooth faces (Liu et al., 2014; Holz et al., 2016), which led to the definition
of the new ice crystal properties in the MODIS collection 6 product (Platnick et al., 2017). Recently, Wang et al. (2021) presented a retrieval using observations of the Airborne Multi-Angle Spectro-Polarimetric Imager (AirMSPI) for thin cirrus over ocean and found 8-element columnar crystals to best represent the observations, with severely roughened surfaces for polarized reflectance measurements and smooth surfaces for the total intensity.

Most in situ observations report ice crystals with more rough surfaces and complex rather than pristine shapes: Schnaiter
et al. (2016) and Järvinen et al. (2018) found that most ice crystal shapes are highly complex rather than pristine. Järvinen





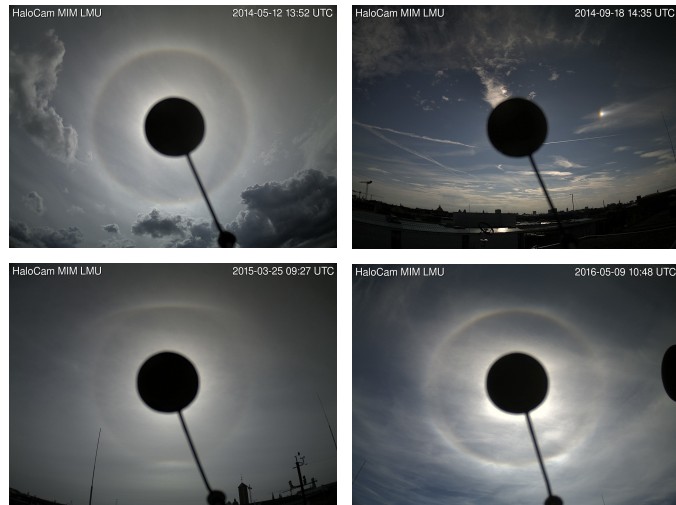

**Figure 1.** Examples of halo displays observed at the Meteorological Institute of the LMU in Munich. The sun is blocked by a black circular shade to avoid stray light and saturation of the camera sensor. Top left: 22° halo. Top right: right-hand 22° parhelia or sundog. Bottom left: faint 22° halo with upper and lower tangent arc. Bottom right: 22° halo with circumscribed halo.

et al. (2018) found from in situ observations using the PHIPS air-craft probe (Abdelmonem et al., 2016; Schnaiter et al., 2018) and Polar Nephelometer (PN) (Gayet et al., 1997; Crepel et al., 1997) that an overwhelming fraction (between 61% and 81%) of atmospheric ice crystals exhibit mesoscopic deformations and could be best represented by a flat and featureless angular scattering function. The probed scattering angle region was 18 to 170° in case of PHIPS and 15 to 162° with a resolution of

3.5° for the PN.

These findings of predominantly rough and complex crystals with featureless scattering phase functions seem to be in disagreement with sightings of halo displays, which form by refraction and reflection of light by smooth hexagonal ice crystals. Forster et al. (2017) showed that at least 25% of all cirrus clouds produce 22° halos, which are only one of the three most common halo types and formed by randomly oriented hexagonal crystals (cf. Fig. 1). In this study we focus on observations in

the scattering angle region of the 22 and 46° halos to shed light on the forward scattering part of the ice crystal phase function containing these halo features. Since this forward scattering angle range is not accessible from satellite-based observations this study adds an important puzzle piece to finding representative ice crystal optical properties for cirrus clouds.

In this study, we investigate a new method to retrieve ice crystal shape and surface roughness from calibrated camera observations of halo displays using HaloCam (Forster et al., 2020). This retrieval method makes use of scattering features,

commonly known as halo displays, which can be observed as bright and colorful circles and arcs in the sky radiance and are caused by details of ice crystal scattering characteristics.

Halo displays are produced by hexagonal ice crystals with smooth faces via refraction and reflection of light as described by Wegener (1925); Greenler (1980); Minnaert (1993) and Tape (1994). Figure 1 illustrates the most frequent halo displays: the 22° halo (top left) is formed by randomly oriented hexagonal ice crystals and appears as a bright ring around the sun at a





scattering angle of about 22°. The top right image in Fig. 1 shows a bright 22° parhelion, commonly called sundog, on the right side of the sun. This type of halo is caused by oriented hexagonal plates. Upper and lower tangent arcs, which are produced by oriented ice crystal columns, are shown on the lower left in Fig. 1. While ice crystal orientation also has significant effects on the global radiative budget (Noel and Sassen, 2005) as well as remote sensing of ice cloud properties, this study focuses

on randomly oriented ice crystals for a start and leaves investigation of oriented crystals for a future study. Halo displays contain valuable information about ice particle size, shape and orientation (Lynch and Schwartz, 1985; Sassen et al., 1994; van Diedenhoven, 2014; Flatau and Draine, 2014). van Diedenhoven (2014) showed that the brightness contrast of the 22° halo in ice crystal scattering phase functions is related to the aspect ratio and surface roughness of the crystals. Quantitative analysis of the frequency of occurrence as well as the brightness contrast of halo displays can therefore help determine ice crystal shape,

surface roughness and orientation in cirrus clouds.

In this study we present a novel method to retrieve ice crystal shape and surface roughness in cirrus clouds using ground-based imaging observations of the 22 and 46° halo scattering angle region. To the authors' knowledge, this is the first quantitative and systematic analysis of a long-term dataset of halo observations. So far, investigations of halo displays regarding ice crystal properties have been limited to qualitative analysis of single case studies (Lynch and Schwartz, 1985). Long-term

studies have focused primarily on the frequency of halo displays with high personnel effort (Sassen et al., 2003).

The paper is structured as follows: Section 2 explains the retrieval method including a detailed description of the ice crystal optical properties, HaloCam observations, and ancillary data used for the retrieval. In section 3 we describe the results of the retrieval applied to a dataset of eight different days between September 2015 and November 2016. The sensitivity of the results to the choice of the aerosol properties and to uncertainties introduced by the camera calibration are investigated. Further

sensitiviy studies of the retrieval and details on ancillary data are provided in the appendix. We close with a discussion of the retrieval results in section 4 and close with summarizing our key findings in section 5.

## 2   Retrieval of ice crystal properties

Cirrus clouds featuring a halo display contain at least a certain amount of smooth hexagonal ice crystals. The frequency of these cirrus clouds, which will be referred to as "halo-producing" cirrus in the following, provides therefore a first estimate of

the minimum fraction of smooth hexagonal ice crystals in cirrus clouds. Forster et al. (2017) estimate from a 2.5 year dataset of HaloCam observations in Munich that about 25% of the cirrus clouds produced a 22° halo. In both the 2017 and the present study we refer to cirrus clouds as non-precipitating ice clouds with a cloud base temperature of $-20\,°C$ or colder.

More detailed information about ice crystal properties can be obtained by analyzing the brightness contrast of the 22° halo and the radiance distribution around the halo. While the brightness contrast of the halo is mostly sensitive to ice crystal shape,

surface roughness, and size, the radiance distribution depends mainly on the cirrus optical thickness (COT). To retrieve all ice crystal properties simultaneously, the radiance measurements of the 22° halo have to be compared with radiative transfer simulations. Look-up tables (LUT) of radiance distributions across the 22° halo were compiled using the libRadtran radiative transfer package (Mayer and Kylling, 2005; Emde et al., 2016) with the DISORT solver (Stamnes et al., 1988) and com-





pared to several days of HaloCam observations to determine the optical and microphysical properties which best represent the observations. The LUT comprises different ice crystal habits, surface roughness values, effective radii, COTs, and AOTs. Furthermore, the LUT is calculated for different SZAs and observation geometries (cf. Tab. 2). For the surface albedo, aerosol type, atmospheric profile and cloud height, fixed parameters were chosen. Finally, the radiance measurements were compared

with the LUT's precomputed radiance distributions to find the best match. The following sections provide more details on the compilation of the LUT and application of the retrieval to long-term HaloCam observations.

## 2.1 Ice crystal shape and roughness models

Optical properties based on Yang et al. (2013) (referred to as YG13 in the following) were used for eight different habits: solid columns, hollow columns, plates, 8-element columns, 5-element plates, 10-element plates, solid bullet rosettes, and hollow

bullet rosettes, all of which are based on hexagonal crystal symmetry. Droxtals were not considered for the retrieval since they do not produce a 22° halo (Yang et al., 2013). Since this parameterization provides only three different roughness levels (smooth, moderately roughened, and severely roughened), the optical properties of smooth and severely roughened ice crystals were mixed linearly to achieve a continuous distribution of roughness levels. For each habit separately, optical properties of smooth and rough ice crystals were mixed by scaling their extinction coefficients in the radiative transfer simulations. The

smooth crystal fraction

$$\mathrm{SCF} = \beta_{\mathrm{ext,smooth}}/\beta_{\mathrm{ext,total}}\,, \tag{1}$$

with $\beta_{\mathrm{ext,total}} = \beta_{\mathrm{ext,smooth}} + \beta_{\mathrm{ext,rough}}$ ranges between $0 \leq \mathrm{SCF} \leq 1$, resulting in a rough crystal fraction of

$$\mathrm{RCF} = 1 - \mathrm{SCF} = \beta_{\mathrm{ext,rough}}/\beta_{\mathrm{ext,total}}\,. \tag{2}$$

This approach is inspired by the findings of Schmitt and Heymsfield (2014) and Liu et al. (2014) who suggest to separate ice

crystals into simple and complex particles.

The radiative transfer simulations for compiling the LUT were performed using the U.S. standard atmospheric profile Anderson et al. (1986) and assuming a cirrus cloud between $10\,\mathrm{km}$ to $11\,\mathrm{km}$ height. Sensitivity studies in Appendix B show that the effect of cloud base height, geometric thickness as well as atmospheric profile cause a bias in the 22° halo radiances of less than 1%. Furthermore, to save computation time, the radiative transfer simulations were performed for a representative

wavelength of each color channel of HaloCam$_{\mathrm{RAW}}$ rather than integrating over the spectral response function: $618\,\mathrm{nm}$ for the red, $553\,\mathrm{nm}$ for the green, and $498\,\mathrm{nm}$ for the blue channel. Figure B3 in Appendix B shows that this causes a bias of 1.5% for the blue, 2.0% for the green, and 1.2% for the red channel.

## 2.2 HaloCam observations and ancillary data

To obtain representative results for ice crystal properties of halo-producing cirrus clouds, long-term observations are required.

These are provided by the weather-proof sun-tracking camera HaloCam$_{\mathrm{RAW}}$ was installed in September 2015 on the rooftop platform of MIM (Forster et al., 2020). Between 22 September 2015 and 31 December 2016 HaloCam$_{\mathrm{RAW}}$ recorded scenes





with a 22° halo on 52 days with a temporal resolution of $10\,\text{s}$. The automated halo detection algorithm HaloForest, described in Forster et al. (2017), was used to filter the HaloCam$_{\text{RAW}}$ images for 22° halos. Additional sun photometer measurements are used to constrain aerosol optical thickness (AOT) and cirrus optical thickness (COT). As demonstrated in Appendix B, additional knowledge about these two parameters is critical to retrieve information about the ice crystal microphysical properties.

5 The aerosol optical thickness was derived from the AERONET AOT product (Holben et al., 1998) (version 2) for the observation site on the MIM rooftop platform. The AOT during the time of the halo observation is constrained to a $2\sigma$ confidence interval around the daily average AOT estimated during clearsky periods.

 The cirrus optical thickness (COT) introduces an ambiguity in the brightness contrast of the 22° halo (Forster et al., 2017). Constraining the COT is therefore necessary for a stable retrieval. For this retrieval the COT is derived from sun photometer

10 measurements using the SSARA instrument (Toledano et al., 2009, 2011). SSARA provides direct sun measurements with a temporal resolution of $2\,\text{s}$ which are much more suitable for the observation of the highly variable cirrus clouds than AERONET with $15\,\text{min}$ (Holben et al., 1998). The COT is derived by calculating the total optical thickness from the SSARA direct sun measurements. The previously estimated AOT is then subtracted and a correction factor is applied to account for the increased forward scattering of the ice crystals (Reinhardt et al., 2014) (cf. Appendix B1.2). The retrieval is applied to the red channel of

15 HaloCam$_{\text{RAW}}$ with a central wavelength of $618\,\text{nm}$ (cf. Fig. B3) to minimize the relative contribution of Rayleigh and aerosol scattering compared to the scattering by ice crystals.

**Table 1.** HaloCam$_{\text{RAW}}$ 22° halo days between 22 September 2015 and 31 December 2016.

| Date | Start time | End time | No. of images |
|---|---|---|---|
| 2015-09-22 | 6:38 UTC | 11:14 UTC | 1054 |
| 2015-11-08 | 10:00 UTC | 10:37 UTC | 198 |
| 2015-11-10 | 9:00 UTC | 10:23 UTC | 88 |
| 2016-01-20 | 9:36 UTC | 11:37 UTC | 544 |
| 2016-02-02 | 8:00 UTC | 14:00 UTC | 1029 |
| 2016-02-06 | 12:00 UTC | 15:20 UTC | 724 |
| 2016-04-21 | 11:34 UTC | 13:52 UTC | 770 |
| 2016-11-04 | 10:27 UTC | 10:40 UTC | 78 |
| Total | | | 4400 |

 Using additional observations of AOT requires clear-sky periods before and/or after the 22° halo event. Simultaneous AOT and COT observations from SSARA and AERONET (including the necessary clear-sky scenes) together with 22° halo observations from HaloCam$_{\text{RAW}}$ are available for only 8 of the 52 days listed in Table 1. Figure 2 shows an example of the AOT and

20 apparent COT derived from sun photometer measurements on 21 April 2016. The AOT is obtained from the AERONET dataset and is represented by turquoise stars. The daily average AOT amounts to about $0.08 \pm 0.04$ at $618\,\text{nm}$. The blue dots in Fig. 2 indicate the apparent COT derived from SSARA direct sun measurements, which are available from about 11:30 UTC. The

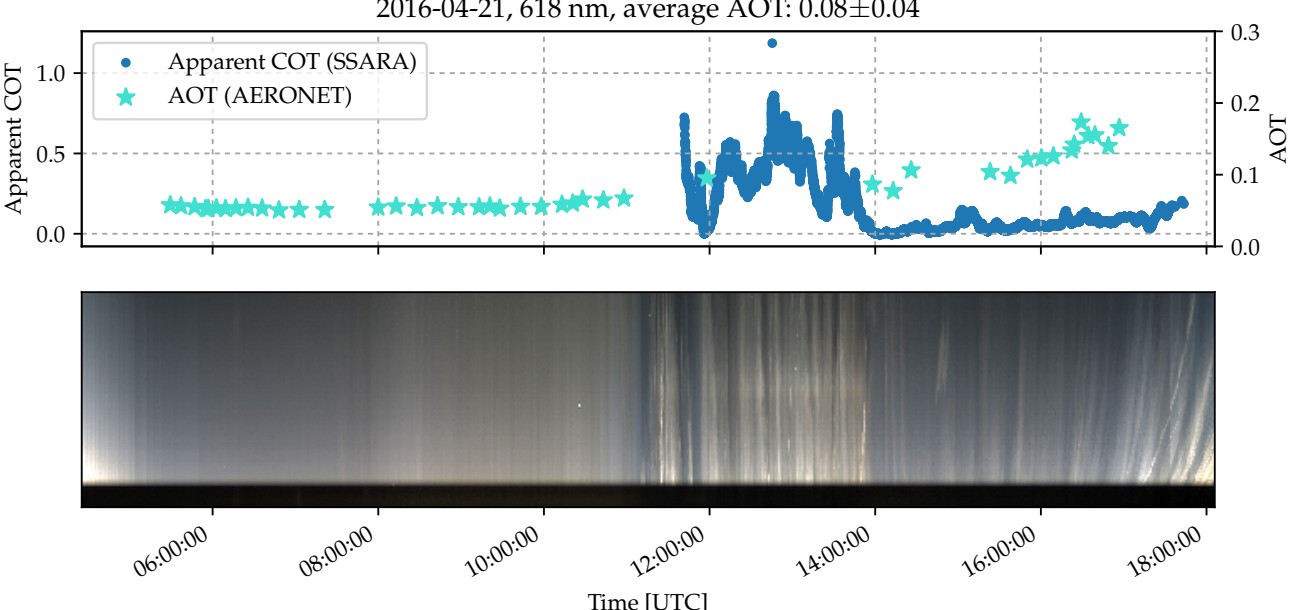

**Figure 2.** Top: AERONET aerosol optical thickness (AOT, turquoise stars) and apparent cirrus optical thickness (COT) derived from SSARA measurements (blue dots) for a wavelength of $618\,\mathrm{nm}$. Bottom: time line of HaloCam pixel slices above the sun.

lower panel of Fig. 2 shows slices of the HaloCam images along the principal plane above the sun. 22° halos and upper tangent arcs appear as a bright line in the center of the panel with a reddish inner, i.e. lower, edge from about 11:30 until 14:00 UTC.

The HaloCam$_{\mathrm{RAW}}$ observations were geometrically and radiometrically calibrated as described in Forster et al. (2020). For applying the retrieval to the HaloCam$_{\mathrm{RAW}}$ data the LUT was calculated for a wavelength of $618\,\mathrm{nm}$ with a surface albedo of

5   0.065. The remaining LUT parameters are provided in Table 2. To use as much information as possible from the HaloCam$_{\mathrm{RAW}}$ images for the retrieval, the radiative transfer simulations for the LUT were performed for the viewing angles of all five image segments. The file size of the LUT and observations was then reduced by averaging both simulated and measured images over the five segments in the direction of the relative azimuth angle $\varphi$ (Forster et al., 2020, Fig. 3b). Thus, a separate LUT was compiled for each of the five HaloCam$_{\mathrm{RAW}}$ image segments which are evaluated separately.

10   ## 3   Application and retrieval results

The retrieval was performed as follows: for each HaloCam$_{\mathrm{RAW}}$ image the LUT was interpolated to the respective SZA and constrained to the AERONET daily mean AOT within a $2\sigma$ confidence interval. For each HaloCam$_{\mathrm{RAW}}$ image time stamp, the LUT was also constrained to SSARA's COT measurements within a $2\sigma$ confidence interval averaged over a $\pm 5\,\mathrm{min}$ time interval. This 10 min window was chosen to account for the slightly different pointing directions $\Theta = 0°$ (sun photometer) and $\Theta = 22°$

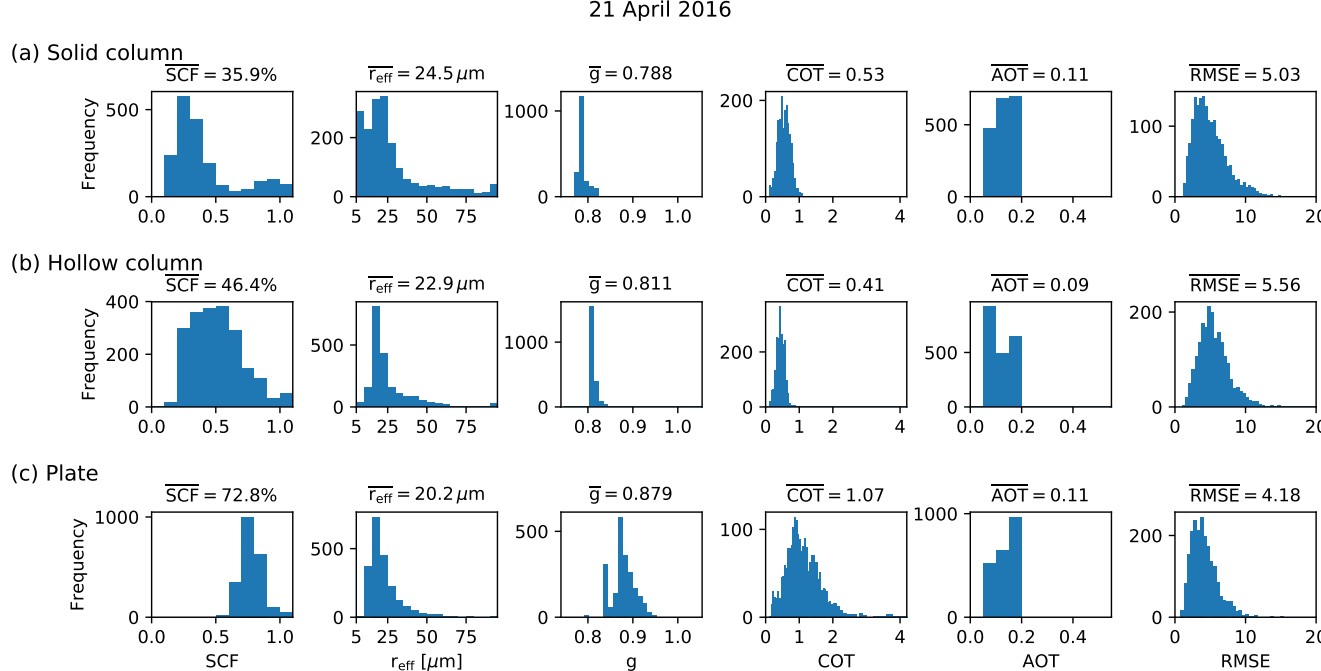

**Figure 3.** Retrieval results for 21 April 2016 for three selected YG13 ice crystal shapes: (a) solid columns, (b) hollow columns, and (c) plates. The different panels show histograms of the best matching LUT parameters for the smooth crystal fraction (SCF), effective radius ($r_{\mathrm{eff}}$), asymmetry factor (g), cirrus optical thickness (COT), aerosol optical thickness (AOT), and the RMSE between LUT and measurement (from left to right). The results were filtered for a halo ratio $\mathrm{HR} > 1$ to ensure that only image slices with $22°$ halo were analyzed and the uppermost image segment was excluded from the analysis to avoid applying the retrieval to upper tangent arcs.

(halo display) in combination with the unknown wind direction. The COT was derived from SSARA's total optical thickness observations by subtracting AERONET AOT and correcting the resulting apparent COT for the enhanced forward scattering by ice crystals according to Eq. B5. Then, each of the five averaged radiance distributions measured with HaloCam$_{\mathrm{RAW}}$ was compared to the LUT elements with the respective viewing geometry. Measurements and LUT are compared via the root mean

5    squared error (RMSE), which is calculated by

$$\mathrm{RMSE} = \sqrt{\sum_{i=1}^{n} \frac{L_{\mathrm{meas,i}} - L_{\mathrm{LUT,i}}}{n}} \,. \tag{3}$$

using the measurements $L_{\mathrm{meas,i}}$ and LUT elements $L_{\mathrm{LUT,i}}$ within the considered scattering angle range and averaged over the number of elements $n$. All LUT elements with a RMSE within the $2\sigma$ measurement uncertainty, averaged over the scattering angle range, are considered possible solutions for the cirrus optical and microphysical properties

10    $$\mathrm{RMSE} \leq \sum_{i=1}^{n} \frac{2\sigma_{\mathrm{L,meas,i}}}{n} \,. \tag{4}$$



**Table 2.** LUT parameters: minimum, maximum value and resolution for smooth crystal fraction (SCF), effective radius ($r_{\mathrm{eff}}$), cirrus optical thickness (COT), aerosol optical thickness (AOT), and solar zenith angle (SZA). COT and AOT are defined at a wavelength of $550\,\mathrm{nm}$.

| LUT parameter | min | max | resolution |
| --- | --- | --- | --- |
| SCF | 0% | 100% | 5% |
| $r_{\mathrm{eff}}$ | $5\,\mu\mathrm{m}$ | $90\,\mu\mathrm{m}$ | $5\,\mu\mathrm{m}$ |
| COT | 0.1 | 2.0 | 0.05 |
| | 2.1 | 3.0 | 0.1 |
| | 3.2 | 4.0 | 0.2 |
| AOT | 0.00 | 0.50 | 0.05 |
| SZA | 25° | 30° | 5° |
| | 40° | 70° | 10° |

The LUT element with the minimum RMSE represents the best match.

## 3.1 Using information of the 22° halo

In this first step, observations and LUT are compared in the scattering angle range between $18° \leq \Theta \leq 25°$ with an angular resolution of $0.5°$. Maximizing the scattering angle range, which is used for this comparison, provides more information. On the other hand, for an increasing angular region inhomogeneities in the cirrus optical and microphysical properties become relevant. The goal of the retrieval is to find the ice crystal properties which best match the observed radiance distributions across the scattering angle range of the 22° halo. Therefore, the scattering angle range was optimized to cover as much as possible of the vicinity of the 22° halo in addition to its peak while keeping it as small as possible to avoid inhomogeneities of the cirrus cloud.

To ensure that only samples with a clearly visible 22° halo are considered, the results were filtered for a halo ratio $\mathrm{HR} > 1$ (cf. Eq. 1 in Forster et al. (2017)) and the uppermost image segment (no. 3) was excluded. This segment might contain signatures of the upper tangent arc, which is produced by oriented ice crystal columns. Sundogs appear in the left and right image segments (no. 1 and 5) only for $\mathrm{SZA} < 45°$ at scattering angles of $\Theta > 29°$ which does not interfere with the 22° halo (as discussed in Forster et al. (2017)). While observations of upper tangent arcs and sundogs contain valuable information about the fraction of oriented columns and plates, they are excluded for the retrieval presented in this study which focuses on randomly oriented crystals.

Figure 3 presents the retrieval results for the 3080 samples (four segments per image) of a 22° halo observed on 21 April 2016. The histograms display the retrieved LUT parameters based on the assumption that the representative ice crystal habit either is solid columns (a), hollow columns (b), or plates (c). The retrieved values for the SCF, effective radius, asymmetry




factor $g$, COT, and AOT are provided as histograms with parameter boundaries and bins as defined in the LUT. The RMSE between LUT and measurement is provided in the rightmost panels of Fig. 3.

For solid columns (Fig. 3a) the SCF peaks below 50% with HaloCam$_{RAW}$'s 22° halo observations are represented best by a mean SCF of 35.9% and RCF of 64.1% (cf. Eq. 2). The ice crystal effective radii peak at 20 µm with a mean value of 24.5 µm and a mean asymmetry factor of 0.788. The majority of COT values are below 1 with a mean value of 0.53, whereas the AOT (constrained between 0.05 and 0.15 using AERONET data) yields a mean value of 0.11. In the case of hollow columns (Fig. 3b), the retrieved SCF ranges around 50% with effective radii of 22.9 µm on average and a mean asymmetry factor of about 0.811. The average COT of 0.41 is slightly smaller compared to the solid column case. For ice crystal plates (Fig. 3c) a larger SCF of about 72.8% (and RCF of 27.2%) on average is required to match the brightness contrast of the 22° halo measured with HaloCam$_{RAW}$. The mean effective radius with 20.2 µm is slightly smaller compared to the solid column case. Assuming plates as dominating ice crystal habit causes a larger asymmetry factor on average with 0.879. The average COT amounts to 1.07 with a few values larger than 2. The retrieved COT in case of plates is significantly larger compared to solid and hollow columns, due to the increased forward scattering indicated by the large asymmetry factors. Increasing the asymmetry factor (i.e. the amount of forward scattering) of ice crystals in a cloud with constant crystal concentration would result in higher radiances values measured by the same detector (cf. Appendix B1.2). Compared with solid and hollow columns, the plate habit shows the smallest RMSE values for this dataset.

Figure 4 displays the results of the retrieval applied to the eight days of 22° halo observations with HaloCam$_{RAW}$. The upper panel presents the retrieved SCF for each day and for the eight habits. By grouping the ice crystal habits into columnar (green), hollow (pink), and plate-shaped (blue) crystals, the average SCF clusters at ~30%, ~60%, and ~80%, respectively. A similar clustering results for the asymmetry factor, which is smallest for columnar crystals and largest for plate-like crystals. In contrast to the differences of the retrieved mean SCFs and asymmetry factors among the habits, the retrieved mean effective radii, shown in the third panel of Fig. 4, seem to be almost independent of ice crystal habit and roughness. This confirms that the width of the 22° halo is primarily determined by ice crystal size, while shape and surface roughness play a minor role. The mean effective radius amounts to about 20 µm. Due to the skewed distribution of the retrieved effective radii (cf. Fig. 3), more than 90% of the results are smaller than 40 µm.

Figure 5 (upper panel) shows the cloud top (circles) and base (dots) height represented by the mean value and standard deviation, which were derived from co-located measurements of the MIRA-35 cloud radar (Görsdorf et al., 2015) on the MIM rooftop platform. On 4 November 2016 cirrus clouds formed only in the South and South-East during the 22° halo event (cf. Table 1). Thus, the zenith-pointing cloud radar did not detect the cirrus cloud observed by HaloCam$_{RAW}$ and therefore no cloud height could be provided. In the other cases the cirrus clouds had a larger horizontal extent and the 22° halo was visible over a longer time period. The cloud top height varied around 10 km except for 20 January 2016 with 6 km. The cloud base height exhibits a larger variability between 5 km and 10 km. The corresponding temperatures at cloud top (circles) and cloud base (dots), indicated by mean value and standard deviation, are displayed in the lower panel of Fig. 5. The threshold temperature for homogeneous nucleation is represented by the blue dashed line at −38 °C. For all seven cases the cloud top temperature was equal or colder than −38 °C while the cloud base temperature varied between −10 °C and −50 °C on average. It is noteworthy



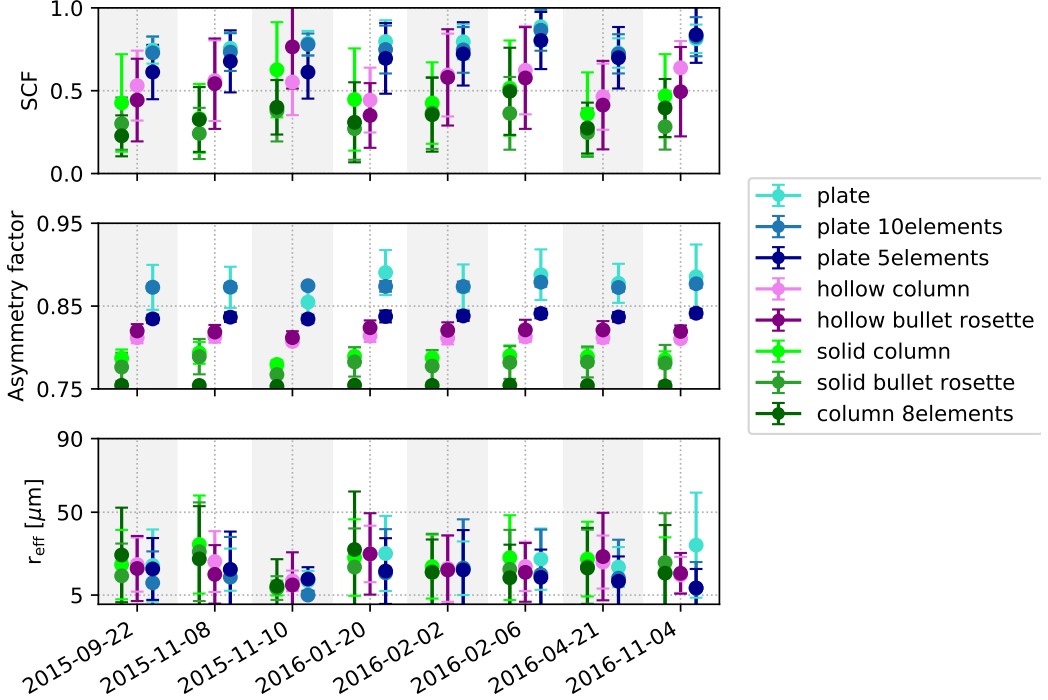

**Figure 4.** Retrieval results for all eight days listed in Table 1 and for all eight crystal shapes of the YG13 optical property database. Results are shown for the SCF (top), asymmetry factor (center), and effective radius (bottom) using the mean value within a $1\sigma$ confidence interval. Note that the underlying distributions might be skewed as depicted in Fig. 3. Blue, pink and green color tones are used to group the ice crystal shapes into plate-like, hollow, and columnar shapes, respectively. Darker colors indicate more complex crystals.

that the coldest and thinnest cirrus on 10 November 2015 with a cloud base temperature of about $-50\,°\text{C}$ coincides with the smallest retrieved effective radii in Fig. 4. This tendency is in agreement with (e.g. Bailey and Hallett, 2009; Baran, 2012) who report the smallest ice crystals close to cloud top and at the coldest temperatures in case of synoptic cirrus. While observing ice crystals directly at cloud top is impossible for ground-based imaging in case of thick clouds, geometrically thin cirrus

5  provide an opportunity to infer ice crystal properties close enough to cloud top and ensure more homogeneous atmospheric conditions, which are conducive for a more homogeneous size and shape distribution. van Diedenhoven (2021) also found effective radii of ice crystals to decrease with increasing cloud top height and thus decreasing temperature using the airborne Research Scanning Polarimeter's (RSP) observations together with reanalysis data from Goddard Earth Observing System Model Forward Processing (GEOS-FP) data assimilation system.

10  Table 3 presents the retrieved SCF, effective radius and asymmetry factor for all evaluated days, sorted by increasing mean RMSE. The retrieval revealed that ice crystal plates have the overall smallest mean RMSE and thus seem to match the HaloCam$_{\text{RAW}}$ observations better in the scattering angle range between 18 and $25°$ than the other seven habits of the





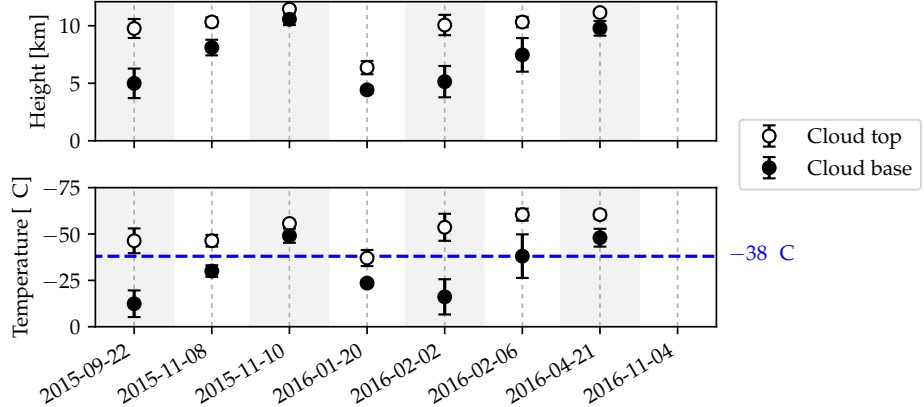

**Figure 5.** Top: cloud top (circles) and base (dots) height was derived from cloud radar observations. Bottom: the corresponding temperature was estimated from radiosonde profiles launched at Oberschleißheim. The dashed blue line indicates the threshold for homogeneous nucleation at a temperature of $-38\,°C$. The results are provided by the mean values within a $1\sigma$ confidence interval over the time periods with a visible $22°$ halo.

**Table 3.** Retrieval results evaluated for all eight days. Mean value and $1\sigma$ standard deviation are provided for the smooth crystal fraction (SCF), effective radius, and asymmetry factor, sorted by increasing mean RMSE.

| Habit | RMSE | SCF [%] | Effective radius [μm] | Asymmetry factor |
| --- | --- | --- | --- | --- |
| plate | 3.73 | $80 \pm 10$ | $21.9 \pm 17.1$ | $0.880 \pm 0.028$ |
| 8-element column | 4.10 | $30 \pm 20$ | $22.6 \pm 24.5$ | $0.752 \pm 0.001$ |
| solid column | 4.14 | $40 \pm 30$ | $23.6 \pm 20.4$ | $0.787 \pm 0.011$ |
| 10-element plate | 4.16 | $70 \pm 10$ | $14.1 \pm 20.9$ | $0.875 \pm 0.005$ |
| 5-element plate | 4.24 | $70 \pm 20$ | $16.4 \pm 16.8$ | $0.837 \pm 0.006$ |
| solid bullet rosette | 4.55 | $30 \pm 20$ | $19.2 \pm 20.8$ | $0.779 \pm 0.020$ |
| hollow column | 4.67 | $50 \pm 20$ | $22.1 \pm 15.2$ | $0.812 \pm 0.007$ |
| hollow bullet rosette | 5.25 | $50 \pm 30$ | $21.8 \pm 21.0$ | $0.821 \pm 0.011$ |

YG13 database. The best matching LUT elements of ice crystal plates have a SCF of $(80 \pm 10)\,\%$, an effective radius of $(21.9 \pm 17.1)\,\mathrm{\mu m}$ and an asymmetry factor of $0.880 \pm 0.028$. With increasing RMSE, the plates are followed by 8-element columns and solid columns. Hollow columns and bullet rosettes have the largest mean RMSE.

In the following we assess how stable the retrieved ice crystal habit is considering the necessary assumptions regarding
5 spectral response, aerosol type, and radiometric uncertainty: Using the representative wavelength of the HaloCam$_{\text{RAW}}$'a red channel instead of accounting for the spectral response of the channel, introduces a small bias of less than 2% for the (cf. Fig. B3). Since the LUT was calculated for the OPAC continental average aerosol type, the retrieval results might be biased if



**Table 4.** Best match habit for the retrieval applied to HaloCam$_{RAW}$ daily observations for the default retrieval (first column) and considering the spectral response (second column), followed by the continental clean, polluted, and urban aerosol type. The habits vary between plates (plate), 5-element plates (5-plate), 10-element plates (10-plate), 8-element columns (8-col), solid columns (sCol), hollow columns (hCol), and solid bullet rosettes (sbRos).

| Date | Default | Sensitivity tests | | |
|---|---|---|---|---|
| | | Spectral response | Aerosol type | |
| | | | contin. clean | contin. polluted | urban |
| 2015-09-22 | 8-col | 8-col | 8-col | 8-col | plate |
| 2015-11-08 | plate | plate | plate | plate | plate |
| 2015-11-10 | sCol | sCol | hCol | sCol | sbRos |
| 2016-01-20 | plate | plate | plate | plate | plate |
| 2016-02-02 | sCol | sCol | sCol | sCol | sCol |
| 2016-02-06 | plate | plate | plate | plate | plate |
| 2016-04-21 | 10-plate | 10-plate | 10-plate | 10-plate | 10-plate |
| 2016-11-04 | plate | 5-plate | plate | plate | plate |

the actual aerosol type differs. To obtain a rough estimate of the sensitivity of the retrieval, it was repeated with a modified LUT to model the effect of these approximations. The LUT was modified by multiplication with a slope which is representative for the amount and the sign of the bias introduced by the approximations. The slope was computed by the ratio between two DIS-ORT simulations using the continental average aerosol type as a reference and continental clean, continental polluted, and urban

5 as modifications. In addition a slope was computed by taking the ratio between a simulation accounting for HaloCam$_{RAW}$'s full spectral response vs. its central wavelength. These slopes were computed for each of the 8 ice crystal habits assuming a representative atmospheric setup: COT = 0.8, AOT = 0.1, and a SCF of 30% for columnar crystals, 60% for hollow column crystals, and 70% for plate-like crystals.

Table 4 shows the results of the best matching habit for each day retrieved with the modified LUT. The best matching habit

10 slightly changed for the different modifications of the LUT but only within the plate-like or column-like crystal groups. The ice crystal plates remain the overall best-matching habit in the considered scattering angle range. Another uncertainty for cloud base temperatures higher than $-38\,°C$ might be the presence of supercooled water droplets, which act similar to rough ice crystals in diminishing the 22° halo as discussed in Appendix B. However, Fig. B2 showed that the presence of water droplets has only a small effect on the retrieved SCF.




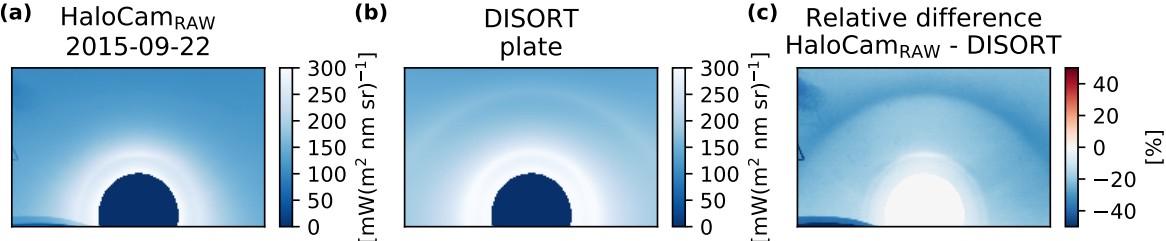

**Figure 6.** Left: HaloCam$_{\mathrm{RAW}}$ R-channel radiance averaged over all 22° halo images for 2015-09-22. Center: DISORT simulations for the best matching ice crystal and cirrus properties in the 22° halo region. Right: relative difference between HaloCam$_{\mathrm{RAW}}$ averaged radiances and the simulated DISORT radiances.

## 3.2  Adding information about the 46° halo

Due to inhomogeneities present in the observed cirrus clouds, we confined the retrieval to the scattering angle range around the 22° halo in a first step. Since the presence or absence of the 46° halo adds important information about the ice crystal shape and surface roughness, in a second step we test here how representative the retrieved LUT parameters are for the whole scene. For this analysis the HaloCam$_{\mathrm{RAW}}$ images were averaged over the whole day and compared with synthetic HaloCam$_{\mathrm{RAW}}$ images simulated with DISORT using the same viewing geometry. Figure 7 shows the averaged HaloCam$_{\mathrm{RAW}}$ images (left) in comparison with DISORT simulations (center). We focused this analysis on 6 of the 8 days, for which the number of halo samples was high and the horizontal extent of the cirrus cloud was large enough to yield homogeneous conditions across both the 22 and 46° halo regions in the averaged image. If ice crystals in the cirrus cloud were able to form a 46° halo, we would expect to see it in the averaged image. Figure 6 displays the comparison with DISORT simulations using ice crystal plates, which were found to best match the observations in the region of the 22° halo (cf. Table 3). Fig. 6a shows the averaged HaloCam$_{\mathrm{RAW}}$ image for 22 September 2015, Fig. 6b the synthetic DISORT image, and Fig. 6c the relative difference between both images in percent. Apparently, the averaged HaloCam$_{\mathrm{RAW}}$ image does not show any 46° halo, whereas the optical properties of plates produce a pronounced 46° halo in addition to the 22° halo in the simulated DISORT image. This can be observed for all evaluated days and is presented here for 22 September 2015 as an example.

Comparing the retrieval results for all eight habits revealed that 8-element columns best match the whole scene of the HaloCam$_{\mathrm{RAW}}$ images – both in the scattering angle range of the 22 and 46° halo. The results using the overall best matching habit of 8-element columns for the DISORT simulations are displayed in Fig. 7. This analysis demonstrates that the scattering angle range around the 46° halo can provide further constraints for the retrieval of ice crystal optical properties.

The retrieval was repeated for the individual HaloCam$_{\mathrm{RAW}}$ images excluding all LUT elements with a 46° halo. The results of the SCF, effective radius and asymmetry factor, averaged over all habits, did not change significantly. In this case, the best matching habit (with the overall smallest RMSE) in the scattering angle region between $18° \leq \Theta \leq 25°$ is the 8-element column followed by solid column. For the whole scene of the HaloCam$_{\mathrm{RAW}}$ images, 8-element columns also proved to slightly better



**Figure 7.** Same as Fig. 6, for selected days and for 8-element columns, the overall best matching ice crystal habit.

represent the observations than solid columns. In YG13's definition 8-element columns are aggregates of eight individual solid columns, each with a slightly different aspect ratio. Since optical properties of aggregates are very similar to those of their individual components, a slight variation of ARs (either as single particles or aggregates) apparently better matches realistic ice crystal populations in cirrus clouds.





It revealed that ice crystal plates match the observations only for larger effective radii of about $50\,\mu\text{m}$ on average. This can be explained with the relationship between ice crystal aspect ratio (AR) and size for the YG13 optical properties: small ice crystal plates have ARs of $\approx 1$ which are effective for the formation of $46°$ halos (van Diedenhoven, 2014) since the ray paths responsible for the 46 and $22°$ halo are both equally likely. Since the overall mean effective radius for all habits except for

plates did not change significantly compared to the results in Fig. 4, it seems that the size-AR parameterization of the plate habit does not represent well the observations. It is important to highlight that the sundogs visible on 8 November 2015 and 6 February 2016 are a clear indication for the presence of oriented ice crystal plates. Together with the missing $46°$ halo, which is produced by randomly oriented ice crystal plates, this could be explained by: (1) The cirrus cloud consists of ice crystal plates too large to be randomly oriented and smaller columnar crystals which are randomly oriented and form the $22°$ halo. (2)

The ice crystal plates have defects on their basal faces, strong enough to inhibit the ray path responsible for the $46°$ halo.

## 4 Discussion

In the following the results of this study results will be further discussed and compared with the literature. Previous studies using passive remote sensing have retrieved quantitative information about ice crystal microphysics primarily from space. Space-borne imaging of optically thin clouds over land is challenging since the measured reflectances are very sensitive to

the surface albedo. While the BRDF is well-known over ocean, it is highly variable over land surfaces. Thus, over land the majority of ice crystal shape and roughness retrievals based on passive remote sensing techniques focuses on optically thicker ice clouds. Moreover, space-borne observations of ice clouds might also include the ice phase of (deep) convection, e.g. anvils of thunderstorms. Ground-based remote sensing of halo displays focuses on rather thin ice clouds instead with a cirrus optical thickness (COT) smaller than about 5 (Gedzelman and Vollmer, 2008). It should also be kept in mind that the results of this

study were obtained from local measurements in Munich in contrast to the space-borne observations which have a global coverage.

### 4.1 Ice crystal shape

This study revealed that the overall best matching ice crystal habits are 8-element and solid columns with a smooth crystal fraction (SCF) of $(30 \pm 20)\,\%$ and $(40 \pm 30)\,\%$ and asymmetry factors at $618\,\text{nm}$ between 0.752 and 0.787, respectively. The

optical properties of solid columns and aggregates of columns, as YG13's 8-element columns, are very similar. The main difference is that the 8-element column is an aggregate of individual solid columns with different aspect ratios and sizes. The fact that 8-element columns proved to be a slightly better match to the observations than solid columns indicates that a distribution of variable aspect ratios and sizes is more representative of the observed cirrus clouds than a "mono-disperse" ice crystal distribution of solid columns. It is noteworthy that 8-element columns, the overall best matching ice crystal habit in this

work, is the same habit as used in the MODIS Collection 6 data product for operational retrieval of ice cloud optical thickness and effective radius (Platnick et al., 2017). While the 8-element columns used for MODIS C6 have severely roughened surfaces to achieve consistency with spectral and polarimetric satellite observations (Baran, 2009; Wang et al., 2018), the results of this



study suggest a fraction of about 30% smooth crystals and 70% severely roughened crystal to account for the presence of halo displays.

Ice crystal columns and aggregates of columns were also found by Holz et al. (2016), however without any smooth crystals, resulting in an asymmetry factor of about 0.75 in the mid-visible spectrum. Also Wang et al. (2014) retrieved a mixture with

a dominating fraction of columnar crystals to best match the MODIS and CALIPSO observations over ocean with a SCF of 10% and an asymmetry factor of 0.778 at a wavelength of $0.65\,\mu m$. These retrievals were performed for $COT < 3$ which is comparable to the optical thickness range observed in this work. Moreover Wang et al. (2021) found columnar crystals as most representative ice particle shape using both total and polarized airborne reflectance measurements from AirMSPI for cirrus clouds of optical thickness up to 5.

Several other studies found plate-like or compact ice crystals to better represent the observations than columns, for example McFarlane and Marchand (2008); van Diedenhoven et al. (2012, 2013); Cole et al. (2014). However, these studies focus on optically thick ice clouds, in particular anvil cirrus, with potentially very different formation mechanisms compared to thin halo-producing ice clouds. Um et al. (2015) studied aspect ratios of natural ice crystals, which were collected during field campaigns by a cloud particle imager, for temperatures between $0\,°C$ and $-87\,°C$ and found that synoptic cirrus is dominated by columnar

crystals, while anvil cirrus contains a larger fraction of plate-like crystals. All evaluated HaloCam$_{RAW}$ observations showed synoptic cirrus or contrail cirrus and did not contain any anvil cirrus. Columnar ice crystals were found to best match these HaloCam$_{RAW}$ observations, which is in agreement with the findings of Um et al. (2015).

Ice crystal plates of the YG13 database produce a pronounced $46°$ halo for the retrieved effective radii which was not visible in the HaloCam$_{RAW}$ observations. YG13 assumes that ice crystal aspect ratio is coupled with crystal size and crystal

top/base faces grow faster than their side faces starting from a cubic shape (aspect ratio 1). For this parameterization, smaller crystals with aspect ratio closer to 1 prouduce more pronounced $46°$ halos compared to larger crystals. A possible explanation why YG13 plates match the HaloCam$_{RAW}$ observations in the $22°$ halo region but not the $46°$ halo region, could be that this parameterization does not represent the observed ice crystal shape: the observed crystals could have larger top/base faces for smaller crystal sizes. Another reason for the missing $46°$ halo could be that crystal base and top faces have defects strong

enough to inhibit the ray path responsible for the formation of this halo type. These defects have been commonly observed in laboratory as well as in situ observations (e.g. Ulanowski et al., 2014) and are not represented in the YG13 database. While YG13's hollow column crystal shape mimics these defects by a cavity at its top and base faces, they appear to be too pronounced to match the HaloCam$_{RAW}$ observations since they introduce a new intensity peak at around $18°$ scattering angle, which is not visible in the observations (Forster et al., 2020, Fig. 13). Optical properties ideally suited for this study would allow choosing

ice crystal size independent of the aspect ratio while still taking into account physical optics effects.

## 4.2   Ice crystal roughness

As shown in Forster et al. (2017), long-term HaloCam observations in Munich revealed that about 25% of the cirrus clouds produced a $22°$ halo. This fraction would be slightly larger when considering other halo types, such as sundogs and upper tangent arcs as well: A visual evaluation of the 6-week HaloCam dataset during the ACCEPT campaign resulted in about 27%



halo-producing cirrus clouds, accounting for all three halo types. The remaining ∼73% of cirrus clouds could either be too opaque (optical thickness >5) for the 22° halo to be visible or contain predominantly rough or complex ice crystals.

The results of the present study focus on cirrus clouds that produce a visible 22° halo. Averaged over all 4400 images, the SCF for columnar, hollow, and plate-shaped crystals amounts to about ∼37%, ∼47%, and ∼73%. Based on the study
by van Diedenhoven (2014) a minimum fraction of smooth crystals of 10% in case of columns or 40% in case of plates can be estimated for the halo-producing cirrus clouds if multiple scattering and scattering by aerosol is neglected. The retrieved fractions in this study taking into account aerosol and cirrus optical thickness result in about 27% (33%) larger SCF for solid columns (plates).

Our finding that columnar ice crystal shapes best represent the HaloCam observations further implies that a majority of rough
ice crystals mixed with a smaller fraction of smooth crystals is sufficient to produce a visible 22° halo. Finding predominantly rough and complex ice crystals to best match the observations is in agreement with the results of several studies based on satellite retrievals. Using multi-angle reflectance measurements, Baran et al. (1998, 1999) and McFarlane and Marchand (2008) found polycrystals and complex crystals to better represent the observations than pristine single crystals. Studies based on multi-angular polarized reflectances from POLDER (Polarization and Directionality of Earth Reflectance) also report that
featureless phase functions, which correspond to roughened or complex crystals, better represent the measurements than phase functions of a single ice crystal habit (Descloitres et al., 1998; Chepfer et al., 2001; Baran et al., 2001; Baran and Labonnote, 2006; Sun et al., 2006). Holz et al. (2016) and Wang et al. (2014) confirmed that rough and complex crystals better match the observations than smooth single crystals for optically thin clouds (COT < 3) using retrievals based on lidar observations and reflectances in the infrared spectrum.

## 4.3  Ice crystal size

The retrieved effective radii in this study are, to the authors' knowledge, the first observational results for 22° halos and yield similar results for all 8 ice crystal habits with 90% of the radii being smaller than $40\,\mu m$ and with a mean value of $20\,\mu m$. Several studies (e.g. Mishchenko and Macke, 1999; Fraser, 1979; Garrett et al., 2007) investigated the size range in which ice crystals produce a 22° halo based on theoretical and analytical considerations for single crystals. A lower boundary for
ice crystal maximum dimensions of about $10\,\mu m$ was found based on an analysis of the 22° and 46° halo in scattering phase functions of Yang and Liou (1996) and Yang et al. (2000). This lower boundary is in agreement with the results from laboratory studies of Sassen and Liou (1979). Another criterion for the formation of a 22° and 46° halo is random orientation. This occurs for compact ice crystals with maximum dimensions smaller than about $100\,\mu m$. Ambiguities might occur since aggregated ice crystals such as bullet rosettes can be oriented while their components are randomly oriented relative to each other (Fraser,
1979; Sassen et al., 1994; Tape, 1994). Another indication for this upper size limit are the findings of Mishchenko and Macke (1997) who report that air bubbles develop in larger ice crystals, which cause the 22° halo to fade. Furthermore, Bailey and Hallett (2002) state that pristine shapes are mostly found in the laboratory for maximum dimensions smaller than about



100 μm. Um and McFarquhar (2015) [1] determined minimum size parameters for the formation of 22° halos as a function of the aspect ratio (AR) resulting in size parameters $\chi = 45$ for compact particles (AR = 1), $\chi = 103$ for plates with AR = 0.1, and $\chi = 182$ for columns with AR = 4. The 46° halo forms starting from size parameters of $\chi = 68$ for plates (AR = 0.5), $\chi = 45$ for compact crystals, and $\chi = 223$ for columns (AR = 2). Unfortunately, these results are difficult to compare to our findings since the effective radius is defined for an ensemble of crystals accounting for different shapes, whereas ice crystal maximum dimension and size parameter are defined for single particles. However, global observations of ice cloud effective radii are available from the MODIS Collection 6 (Yi et al., 2017), which range between 30 μm to 35 μm over land in the northern mid-latitudes. These values are slightly larger than the mean effective radius of about 20 μm we retrieved for ice crystals producing a 22° halo.

## 5   Summary and Conclusions

We present a novel imaging remote sensing method to retrieve ice crystal optical and microphysical properties, with a special focus on ice crystal roughness and shape. Using calibrated RGB images of the automated sun-tracking camera system HaloCam, we exploit the scattering features of the 22 and 46° halo which are formed by randomly oriented hexagonal ice crystals. It can be concluded that the brightness contrast and width of the 22 and 46° halo contain valuable information about ice crystal size, shape, and surface roughness. This retrieval compares measured radiance distributions with look-up tables of radiative transfer (RT) simulations, which were calculated for a range of ice crystal optical properties using the database of Yang et al. (2013) (YG13) and the DISORT RT solver. The YG13 database provides ice crystal optical properties for nine different habits, different sizes, and three levels of surface roughness (smooth, moderately roughened, severely roughened). To achieve continuous roughness levels, the optical properties of smooth and severely roughened ice crystals of a specific habit were mixed linearly with smooth crystal fractions (SCFs) ranging from 0% to 100%. Sensitivity tests showed that if the retrieval is applied to uncalibrated measurements with unknown radiometric response, the retrieved SCF can deviate up to 70% from the true value. If the uncertainty of the radiometric response is smaller than 15%, the error in the retrieved SCF is smaller than about 15%. A reasonable absolute radiometric calibration is therefore required to retrieve quantitative results of the ice crystal properties.

Long-term observations of ice crystal optical and microphysical properties were performed using HaloCam$_{\mathrm{RAW}}$. This camera provides the "raw" signal directly from the sensor and was geometrically and radiometrically calibrated as described in Forster et al. (2020). For the retrieval the red channel was used with an absolute radiometric uncertainty of less than 5%. The machine-learning based image classification algorithm HaloForest (Forster et al., 2017) was used to select HaloCam images with a visible 22° halo. For eight days in total, 22° halo observations with simultaneous sun photometer measurements were

---

[1]Note that the term "circumscribed halo" in Um and McFarquhar (2015) was incorrectly used as a collective term for the 22° and 46° halo. In fact, the circumscribed halo occurs at high solar elevations when upper and lower tangent arcs merge and is formed by oriented columns instead of randomly oriented hexagonal crystals.



available which are used to constrain both cirrus and aerosol optical thickness. The retrieval was applied to a total of 4400 HaloCam$_\text{RAW}$ images and the best matching ice crystal properties were analyzed.

It was found that several ice crystal habits and SCFs match the observations within the averaged measurement error in the scattering angle region around the 22° halo. Plate-like crystals with a large SCF and columnar crystals with a small SCF
could reproduce the same 22° halo within the measurement uncertainty. Averaged over all 4400 images, the SCF for columnar, hollow, and plate-shaped crystals amounts to about ∼37%, ∼47%, and ∼73%. Although ice crystal plates best match the observations in the angular region of the 22° halo, the YG13 optical properties exhibit a pronounced 46° halo for effective radii smaller than about $50\,\mu\text{m}$, which is not visible in the evaluated HaloCam images. Filtering the LUT for elements without a 46° halo yields 8-element columns as best matching ice crystal habit with an average SCF of $(30 \pm 20)\,\%$, an average effective
radius of $(22.6 \pm 24.5)\,\mu\text{m}$ and an asymmetry factor of $0.752 \pm 0.001$. This result is in agreement with satellite-based retrievals for optically thin cirrus which also find aggregates of columns as best matching ice crystal habit (Wang et al., 2014; Holz et al., 2016).

The variation of the retrieved effective radii between the ice crystal habits is much smaller compared to the variation of the SCF and yields an overall mean of about $20\,\mu\text{m}$. The underlying distribution of the retrieved effective radii is skewed towards
smaller values with more than 90% of the radii being smaller than $40\,\mu\text{m}$. Relating the retrieved ice crystal effective radii to the temperature of cloud base and top revealed that the smallest crystals were retrieved for the coldest and thinnest cirrus. This tendency of finding the smallest crystals at the coldest cloud temperatures close to cloud top is in agreement with in situ observations (Baran, 2012).

This study highlights the potential and feasibility of a completely automated method to collect and evaluate halo observa-
tions. Long-term calibrated radiance observations of the 22 and 46° halo scattering angle range together with sun photometer measurements allow the retrieval of ice crystal shape, size, and surface roughness, representative for cirrus clouds. Long-term observations in Munich indicate that about 25% of the cirrus clouds contained about $(50 \pm 30)\,\%$ smooth ice crystals with effective radii of about $(20 \pm 10)\,\mu\text{m}$ regardless of their shape. Accounting for the missing 46° halo in the HaloCam observations, 8-element columns reproduced best the measured radiance distributions across the 22° halo. As a next step, the
retrieval should be applied to all available HaloCam$_\text{RAW}$ observations to date. Filtered for cirrus clouds by using the institute's CLOUDNET product (Illingworth et al., 2007), the retrieval results allow to determine ice crystal habit, SCF, and effective radius representative for cirrus clouds in general – including both halo- and non-halo-producing cirrus.

These observations contribute to an improved understanding of ice crystal optical and microphysical properties. Implemented on different sites, HaloCam in combination with HaloForest can provide a consistent dataset for climatological studies of ice
crystal properties representing optically thin ice clouds: for example anvil cirrus of deep convection in the tropics, or cirrus clouds and diamond dust in high-latitude regions. Representative ice crystal optical properties are required for remote sensing of cirrus clouds as well as climate modeling. To the authors' knowledge, this study presents the first quantitative retrieval for ice crystal shape and surface roughness using ground-based imaging observations of halo displays. Since ground-based observations provide information about the forward scattering part of the ice crystal optical properties, the results of this work
ideally complement the results of satellite-based studies.





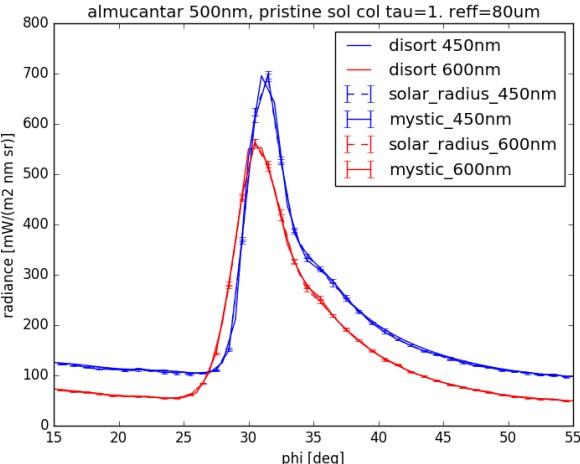

**Figure A1.** Comparison between MYSTIC (lines with error bars) and DISORT (solid lines) for the region of the 22° halo for red (600 nm) and blue light (450 nm). For the MYSTIC simulations, the effect of taking the solar radius into account (dashed lines with error bars) vs. assuming a point source (solid lines with error bars) is shown as well. $10^5$ photons are used for the MYSTIC simulations and 16 streams for DISORT. For all simulations a solar zenith angle of 50° is assumed and the halo slice is computed in the almucantar plane, i.e. with varying azimuth and constant solar zenith angle. The ice cloud was defined with an optical thickness of 1 and consists of smooth solid columns with an effective radius of 80 $\mu$m.

## Appendix A: Radiative transfer simulations of halo displays - DISORT vs MYSTIC

An important choice for creating the look-up tables used in this study is the radiative transfer model. Since cirrus clouds producing visible 22° halos have to be homogeneous across a large scattering angle region (more than 44°), we assume they are horizontally homogeneous and infinitely extended cloud layers. This assumption allows us to use the one-dimensional
radiative transfer solver DISORT (Discrete Ordinate Radiative Transfer) (Stamnes et al., 1988), which is considerably faster for computing radiances compared to MYSTIC, the 3D "Monte Carlo code for phYsically correct tracing of photons In Cloudy atmospheres" (Mayer, 2009; Emde et al., 2010). Using the MYSTIC as a the "physically correct" reference, we tested the performance and accuracy of the DISORT solver for simulating synthetic HaloCam observations of the 22° halo. An important tuning parameter for the Discrete-Ordinate approximation is the number of quadrature angles, also called streams, which must
be large enough to correctly sample the 22° halo. Figure A1 shows the direct comparison of radiances across the 22° halo peak simulated with DISORT (solid curves) and MYSTIC (curves with errorbars). Using 16 streams, DISORT agrees well with MYSTIC within the Monte-Carlo noise for $10^5$ photons considering a $2\sigma$ confidence interval. A second test was performed by taking into account the solar radius instead of assuming a point source (dashed curves with errorbars), which also showed negligible differences compared to DISORT within the $2\sigma$ errorbars. Based on this comparison, we decided to use DISORT
with 16 streams to compute the radiances for the look-up tables.



## Appendix B: Sensitivity studies

In the following the sensitivity of the retrieval on the retrieved smooth crystal fraction (SCF) is tested for different scenarios using the YG13 model for the ice crystal optical properties. LUTs assuming slightly different atmospheric or ice cloud parameters are matched against synthetic measurements simulated with DISORT. The tests are performed for the ice crystal habit,

AOT, the aerosol type, surface albedo, and atmospheric profile. The synthetic measurements were simulated for a wavelength of $500\,\mathrm{nm}$ and a solar zenith angle of $45°$ in the almucantar plane. The SCF is varied between 0 and 1 in steps of 0.05, whereas the cirrus optical thickness ranges between 0.1 and 3. The effective radius, which is related to the width of the $22°$ halo, was demonstrated to be independent from multiple scattering effects. Thus, the sensitivity studies presented in this section focus on the SCF with the effective radius treated as a free parameter, ranging from $10\,\mu\mathrm{m}$ to $90\,\mu\mathrm{m}$ in steps of $10\,\mu\mathrm{m}$. Unless otherwise

stated, ice clouds with different mixtures of smooth and severely roughened solid columns with an aerosol-free atmosphere assuming the U.S. standard atmospheric profile (Anderson et al., 1986) were used for the radiative transfer simulations.

     First, the retrieval error is estimated by applying the retrieval to simulated test cases using LUTs with slight deviations in the assumed atmospheric condition, e. g. surface albedo, AOT, aerosol type. In order to investigate the stability of the retrieval for different ice clouds, simulations were performed for a range of COTs and SCFs for one ice crystal habit population. The

retrieval error is evaluated for the difference between the true and retrieved SCF defined by

$$\Delta\mathrm{SCF} = \mathrm{SCF}_{\mathrm{Retrieved}} - \mathrm{SCF}_{\mathrm{True}}\,. \tag{B1}$$

Figure B1 (Ia) demonstrates the effect of assuming a wrong ice crystal shape. All other LUT parameters are correct. The surface albedo is zero and an aerosol free atmosphere is assumed. The difference of the retrieved smooth crystal fraction is denoted by $\Delta\mathrm{SCF}$. Blue colors indicate an underestimation of the true SCF ($\mathrm{SCF}_{\mathrm{Retrieved}} < \mathrm{SCF}_{\mathrm{True}}$) and red colors represent an

overestimation of the true SCF ($\mathrm{SCF}_{\mathrm{Retrieved}} > \mathrm{SCF}_{\mathrm{True}}$). Calculating the LUT for solid columns and applying it to a cirrus cloud consisting of hollow columns causes a tendency to underestimate the retrieved fraction of smooth ice crystals. This is due to the fact that solid columns produce a brighter halo than hollow columns. Therefore, a smaller fraction of smooth ice crystals is needed in case of the solid columns to produce an equally bright halo. The error of the retrieved fraction of smooth ice crystals is almost independent of the COT but increases with SCF. A maximum error of $\Delta\mathrm{SCF} = -0.45$ occurs for

$\mathrm{COT} = 2.8$ and $\mathrm{SCF} = 0.8$.

     In Fig. B1 (Ib) the sensitivity of the retrieved smooth crystal fraction is tested for an error in the assumed AOT. For this test the surface albedo is set to zero and the "continental clean" aerosol mixture from the OPAC library was chosen. Underestimating the AOT leads to an underestimation of the SCF, especially for very small COTs. The $22°$ halo in the LUT is brighter than in the true data due to the lower AOT, especially for low COTs for which the aerosol scattering features dominate over the halo

features. Therefore, a smaller SCF is sufficient to obtain a $22°$ halo of the same brightness contrast as the true halo. When the COT becomes larger than the AOT the retrieval error tends to decrease. For this test the largest error of the retrieved SCF amounts to $\Delta\mathrm{SCF} = -0.65$ for $\mathrm{COT} = 0.1$ and $\mathrm{SCF} = 0.9$.

     A similar but much less pronounced effect occurs for errors in the assumed aerosol type, demonstrated in Fig. B1 (Ic). For the LUT the "continental polluted" OPAC aerosol optical properties were used whereas the truth is "continental clean" with



**Figure B1.** Sensitivity of the retrieval regarding five different LUT parameters: (a) ice crystal habit, (b) AOT, (c) aerosol type, (d) surface albedo, and (e) atmospheric profile. Three different scenarios were investigated: (I) assuming "perfect" measurements without calibration uncertainty. (II) assuming uncalibrated measurements by treating the radiometric response as free scaling parameter during the retrieval. (III) calibrated measurements with an uncertainty of 15%. To test the sensitivity, a LUT was matched against synthetic measurements simulated with DISORT at a wavelength of $500\,\mathrm{nm}$ and an SZA of $45°$ in the almucantar plane. Synthetic measurements for different COTs and SCFs were calculated and are considered as "truth". The LUTs were calculated for slightly different parameter values or parameterizations for the different tests (a–e) while all other LUT parameters were correct. Panels (a) – (e) show contour plots of the difference between the true and retrieved smooth crystal fraction $\Delta\mathrm{SCF} = \mathrm{SCF}_{\mathrm{Retrieved}} - \mathrm{SCF}_{\mathrm{True}}$. Blue indicates an underestimation ($\mathrm{SCF}_{\mathrm{Retrieved}} < \mathrm{SCF}_{\mathrm{True}}$) and red an overestimation ($\mathrm{SCF}_{\mathrm{Retrieved}} > \mathrm{SCF}_{\mathrm{True}}$) of the true SCF.





a constant AOT of 0.2 and surface albedo zero. In this case the SCF is overestimated for very small COTs. The maximum difference between retrieved and true smooth crystal fraction amounts to $\Delta SCF = -0.3$ for $COT = 0.1$ and $SCF = 0.7$. The results of these two sensitivity studies demonstrate that especially for ground-based remote sensing it is crucial to have an accurate representation of aerosol type and optical thickness in the model setup in order to retrieve information about ice cloud

optical properties. An error in the assumed surface albedo of 0.1 (Fig. B1 (Id)) has a significantly weaker effect on the retrieved smooth crystal fraction with a maximum error of $\Delta SCF = -0.05$ for $COT = 0.1$ and $SCF = 0.1$. For these simulations an aerosol free atmosphere was assumed.

The last sensitivity study shown in Fig. B1 (Ie) investigates the effect of a different atmospheric profile. This results in a slightly different humidity profile which in turn affects the aerosol optical properties. For this experiment the LUT assumes the

U.S. standard atmospheric profile whereas the true profile is the mid-latitude summer atmosphere with higher relative humidity values in the lower layers (Anderson et al., 1986). The results show that for very thin cirrus there is a small difference between true and retrieved smooth crystal fraction of $\Delta SCF = -0.1$ for $COT = 0.1$ and $SCF = 0.8$. In general the introduced error is negligible compared to the errors caused by a wrong representation of the aerosol optical properties.

Figure B1 (II) shows the same sensitivity studies as Fig. B1 (I) but assuming measurements with unknown radiometric

response. To retrieve the best match in the LUT, the radiometric response of the measured radiance is a free parameter. The sensitivity test of assuming a wrong ice crystal shape, shown in Fig. B1 (IIa), yields almost the same results as the study with the calibrated measurements. The underestimation of the SCF is larger for a brighter halo if solid columns are assumed instead of hollow columns with a maximum error of the retrieved SCF of $\Delta SCF = -0.4$ for $COT = 2.2$ and $SCF = 0.85$. Figure B1 (IIb) shows that uncalibrated measurements can lead to large errors of the SCF ranging from an underestimation of

$\Delta SCF = -0.4$ for small COTs to an overestimation up to $\Delta SCF = 0.55$ for $COT > 1$ for an error in the assumed AOT of 0.1. A similar behavior can be observed for the sensitivity test of the aerosol type in Fig. B1 (IIc) which results in a maximum underestimation of the SCF of $\Delta SCF = -0.15$ for small COTs and an overestimation of the SCF up to $\Delta SCF = 0.7$ for $COT = 1.5$ and $SCF = 0.5$. The tendency to underestimate the retrieved SCF for small COTs and a high SCF remains almost the same as for calibrated measurements. The sensitivity studies of the retrieval on wrong assumptions of the surface albedo

(Fig. B1 (IId)) is almost negligible with a maximum error of $\Delta SCF = 0.05$ in the retrieved SCF. An error in the assumed atmospheric profile (Fig. B1 (IIe)) results in a maximum error of the retrieved SCF between $\Delta SCF = -0.35$ and $\Delta SCF = 0.3$ at a COT of 0.1 and 0.9, respectively. This study demonstrates that for uncalibrated measurements the retrieval uncertainties can deviate up to 70% in the retrieved SCF from the errors of the calibrated measurements.

Another test was performed for calibrated measurements with an error of the radiometric response of 15%, which corre-

sponds to the error of HaloCam$_{RAW}$'s R-channel (Forster et al., 2020). Figure B1 (III) shows the results for the same sensitivity studies as in the previous cases (Fig. B1 (I,II)). The results of the ice crystal habit and AOT test in Fig. B1 (IIIa) and Fig. B1 (IIIb) are very similar to the calibrated measurements assuming no error for the radiometric response (cf. Fig. B1 (Ia) and Fig. B1 (Ib)). A slight overestimation of the retrieved SCF occurs for the aerosol type and atmospheric profile test (Fig. B1 (IIIc) and Fig. B1 (IIIe)) compared to the sensitivity of the calibrated measurements assuming no error for the ra-

diometric response. For the aerosol type test (Fig. B1 (IIIc)) the error of the retrieved SCF ranges between $\Delta SCF = -0.15$





and $\Delta$SCF $= 0.15$, whereas for the atmospheric profile test (Fig. B1 (IIIe)) $\Delta$SCF varies between $[-0.35, 0.15]$. The error of the retrieved SCF for the albedo test (Fig. B1 (IIId) is negligible which occurs most likely since errors in the assumed LUT parameters are transferred to the radiometric calibration factor to some extent.

These sensitivity studies demonstrate that the largest retrieval errors occur for wrong assumptions of the ice crystal habit and
the AOT. Thus, for the compiled LUTs all available ice crystal habits for the YG13 optical properties are considered. Under the assumption that the optical properties represent the variability of ice crystals in natural cirrus clouds, the retrieval error for the ice crystal habit is negligible. The AOT is varied in the LUT assuming typical values for Munich. For the remaining LUT parameters, i.e. aerosol type, surface albedo, cloud height, and atmospheric profile, "best guess" fixed values or parameters are chosen. The procedure how the LUT parameters are selected will be presented in the following sections.

Depending on the temperature regime of the cirrus and its evolutionary stage, the cloud can contain supercooled water droplets alongside the ice crystals. Hu et al. (2010) investigated the occurrence frequency, liquid water content, liquid water path, and temperature dependence of supercooled water droplets using global depolarization and backscatter intensity measurements from CALIOP. These observations were combined with temperature information from co-located infrared imaging
radiometer (IIR) and MODIS measurements to derive cloud water paths. This study considers clouds with an optical thickness greater than 0.4. Hu et al. (2010) confirmed the findings of Hogan et al. (2004) who state that supercooled water clouds are rarely found below $-35\,°C$. According to Hu et al. (2010) the probability of water phase occurring in a cloud is almost 0% for $T \leq -35\,°C$ and increases rapidly to almost 100% at about $-10\,°C$.

Since water droplets cannot form halo displays due to their spherical shape, they have in principle a similar smoothing effect
on halo displays as rough ice crystals. Water droplets may therefore not be distinguishable from rough ice crystals by passive ground-based observations in the visible spectral range. To investigate the effect of supercooled water droplets on the retrieved smooth crystal fraction, synthetic measurements were simulated with DISORT for different mixtures of smooth ice crystal columns and water droplets. Similar as for the two-habit LUTs, the fraction of water droplets was increased from 0 for a cloud consisting entirely of smooth solid ice crystal columns to 1 for a pure water cloud. The water cloud optical properties were
calculated with the Mie tool described in Wiscombe (1980). A gamma size distribution $N(r)$ was assumed

$$N(r) = N_0 r^\alpha exp\left(-\frac{r}{r_{\text{eff}}\,\nu_{\text{eff}}}\right),\tag{B2}$$

with the droplet radius $r$, the normalization constant $N_0$, and $\alpha = 7$, which corresponds to an effective variance of $\nu_{\text{eff}} = 1/(\alpha + 3) = 0.1$, as described in Emde et al. (2016). It is assumed that all cloud particles (water droplets and ice crystals) have the same effective radius which was varied between 10 and 90 μm in steps of 10 μm. A LUT assuming different mixtures of
smooth and rough ice crystal columns was matched against these synthetic measurements. The retrieved SCF is displayed in Fig. B2. The error of the retrieved SCF ranges in the interval $\Delta$SCF $\in [-0.1, 0.1]$. This means that water droplets indeed have a very similar effect on the 22° halo as rough ice crystals and introduce an error of the retrieved smooth crystal fraction of $\Delta$SCF $= \pm 0.1$.





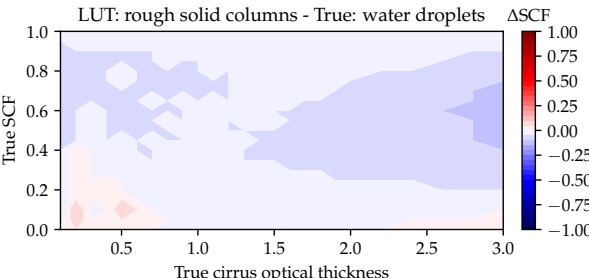

**Figure B2.** Sensitivity studies as in Fig. B1 for measurements assuming mixtures of smooth ice crystal columns with supercooled water droplets.

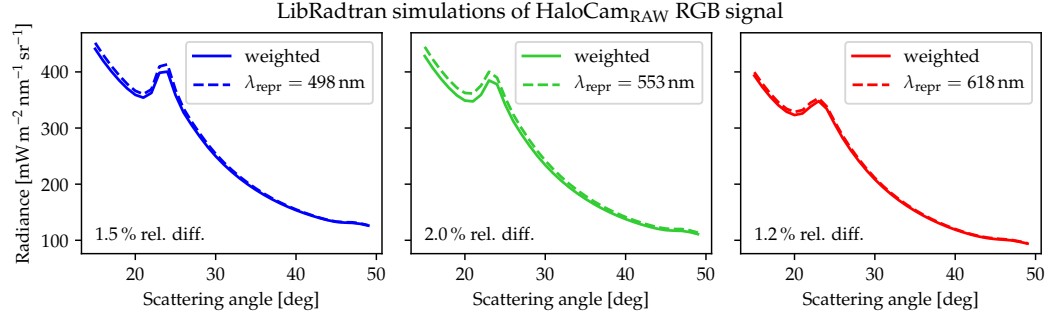

**Figure B3.** Radiative transfer simulations performed with libRadtran for the HaloCam$_{RAW}$ red, green and blue channel in the principal plane above the sun with SZA $= 45°$. A cirrus cloud with COT $= 1$ (at a wavelength $550\,\text{nm}$), $r_{\text{eff}} = 20\,\mu\text{m}$ and a mixture of 25% smooth and 75% severely roughened solid columns was assumed. The continental average aerosol mixture from OPAC was chosen with AOT $= 0.1$ at $550\,\text{nm}$.

The sensitivity of the cloud height and thickness as well as the atmospheric profile on the $22°$ halo radiance distribution was tested. The tests were performed for the HaloCam$_{RAW}$ R-, G-, and B-channel. Varying the cloud base height between $6\,\text{km}$ and $10\,\text{km}$, both with a geometrical thickness of $1\,\text{km}$ resulted in differences of $\ll 1\,\%$. Similar results were obtained for the depth of the cloud which was varied between $1\,\text{km}$ and $4\,\text{km}$. Also the choice of the atmospheric profile is negligible in this spectral

5    range: the difference between a simulations using the U.S. standard atmosphere and the mid-latitude summer atmosphere was $\ll 1\,\%$. Both atmospheric profiles are defined in Anderson et al. (1986).

Furthermore, it was tested whether it is sufficient to perform radiative transfer simulations for a representative wavelength

10    rather than integrating over the full spectral sensitivity curves of HaloCam$_{RAW}$. Figure B3 shows the results of radiative transfer simulations using *libRadtran* for realistic conditions including a cirrus cloud with 25% smooth crystals and a typical AOT of





0.1. The geometry was chosen in the principal plane above the sun (SZA $= 45°$) for scattering angles between $10°$ and $50°$. The solid lines represent spectral simulations integrated over the spectral sensitivity functions for the red, green and blue channel of HaloCam$_\mathrm{RAW}$. The dashed lines display the same simulations but for only one wavelength which is equal to the weighted average of the respective camera channels. The averaged relative differences are overall smaller than 2%. Considering the large

uncertainties of the unknown aerosol type let alone the variability of the ice crystal shape, this uncertainty is considered small enough to allow for monochromatic radiative transfer simulations using the representative wavelengths of each camera channel. The representative wavelengths for HaloCam$_\mathrm{RAW}$ were determined by the weighted average over the spectral response of each channel (Forster et al., 2020) resulting in $618\,\mathrm{nm}$ for the red, $553\,\mathrm{nm}$ for the green, and $498\,\mathrm{nm}$ for the blue channel.

## B1    Ancillary data

The sensitivity studies in Appendix B reveal that the retrieval is influenced by additional parameters. Besides the ice crystal shape itself, the cirrus optical thickness has the strongest impact on the retrieval followed by the aerosol optical thickness and the surface albedo. The subsequent sections present ancillary data which are used to constrain these additional parameters in the retrieval and explain the methods which are used to determine these parameters.

### B1.1    Aerosol optical thickness

According to the study of Schnell (2014) typical aerosol optical thickness (AOT) values for Munich during the period from 2007 to 2010 amount to $0.269 \pm 0.014$ based on AERONET data for a wavelength of $500\,\mathrm{nm}$ and ranged between $0.12 - 0.17$ at $532\,\mathrm{nm}$ for measurements with the Multichannel Lidar System (MULIS) (Freudenthaler et al., 2009). For the period between September 2015 and December 2016 the AERONET AOT at $500\,\mathrm{nm}$ amounts to 0.19 on average as displayed in Fig. B4. 3% of the values range between $0.7 \leq \mathrm{AOT} \leq 3$, possibly due to contamination of very homogeneous cloud layers which are not

filtered out by the AERONET cloud-screening algorithm. To cover the most frequently observed values for Munich, which are displayed in Fig. B4b, the LUT was calculated for AOTs ranging between 0.0 and 0.5 in steps of 0.05. Schnell (2014) also studied the typical aerosol type over Munich using CALIPSO data. Evaluated in geometrical layer depth, the dominant aerosol type was smoke, followed by polluted dust. In fall the continental clean aerosol type was the second largest fraction. Other observed aerosol types were dust and continental polluted aerosol. Unless otherwise stated, the LUT simulations were

performed using the "continental average" mixture which is part of the OPAC database Hess et al. (1998). To constrain the AOT in the retrieval, the daily mean value from AERONET was used within a $2\sigma$ confidence interval.

### B1.2    Cirrus optical thickness

To constrain the cirrus optical thickness (COT) in the retrieval, we make use of the SSARA sun photometer, which is located on the institute's rooftop and provides a high temporal resolution of 2 s. After deriving the total optical thickness from SSARA's

direct sun measurements at a wavelength of 500 nm, we subtract the AERONET AOT from the previous clear-sky scene, interpolated to the cirrus time stamp, to obtain the apparent COT. Due to the enhanced forward scattering in case of ice





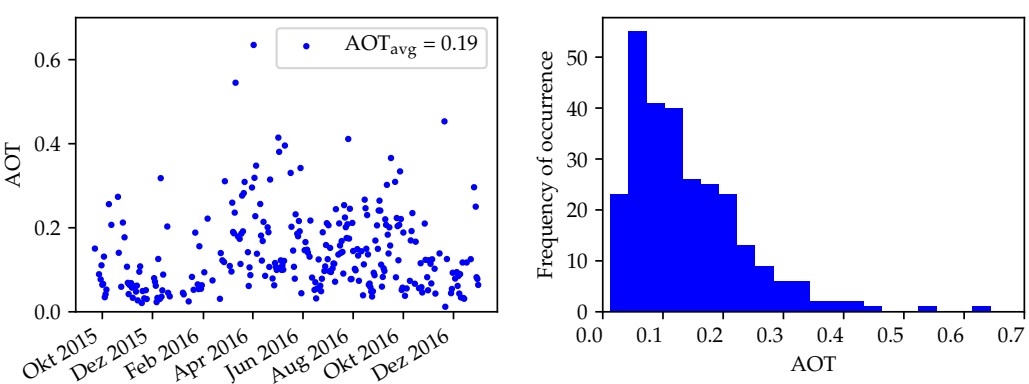

**Figure B4.** AERONET (version 2, level 1.5) AOT at $500\,\mathrm{nm}$ wavelength for the period between September 2015 and December 2016 with an average AOT of 0.19 (left panel). The histogram on the right shows that the most frequent AOTs range between 0 and 0.5 with only 3% of the values between $0.7 \leq \mathrm{AOT} \leq 3$.

crystals, the sun photometer detects a higher signal within its field of view (FOV) for the same concentration of scattering ice compared to aerosol particles, hence the term "apparent" COT. This additional forward scattering contribution can be corrected for by using radiative transfer simulations to compute and tabulate the correction factor $k$.

Similar to the procedure presented in Reinhardt et al. (2014) the concept of the apparent optical thickness is used as in
Shiobara and Asano (1994), Guerrero-Rascado et al. (2013), and Segal-Rosenheimer et al. (2013). According to Bouguer-Lambert-Beer's law, the solar radiance $L$ transmitted by the atmosphere with a slant path optical thickness $\tau_{\mathrm{s}}$ (equivalent to COT in our case) can be denoted as

$$L = L_0 \exp(-\tau_{\mathrm{s}}),$$

where $L_0$ is the solar radiance at the top of the atmosphere. Any detector with a finite FOV which is pointing towards the
10 sun will measure both the direct solar radiance and the diffuse radiance produced by scattering particles and molecules in the atmosphere. The total radiance entering the instrument FOV can be considered as an apparent radiance $L'$ representing the direct and the diffuse part together. The apparent radiance is defined as

$$L' = L_0 \exp(-\tau_{\mathrm{app}}) = L_0 \exp(-k\,\tau_{\mathrm{s}}), \tag{B3}$$

with the apparent optical thickness $\tau_{\mathrm{app}}$ (equivalent to apparent COT in our case). The apparent optical thickness can be related
with the slant-path optical thickness $\tau_{\mathrm{s}}$ by introducing the correction factor $k$

$$\tau_{\mathrm{app}} = k\,\tau_{\mathrm{s}}, \tag{B4}$$





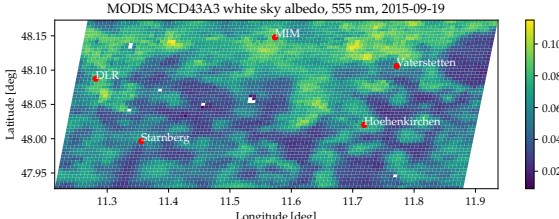

**Figure B5.** MODIS MCD43A3 white sky albedo from 19 September 2015 at a wavelength of $555\,\mathrm{nm}$ displayed for the geographic region which is covered by the projected $22°$ halo between sunrise and sunset throughout the year. The Meteorological Institute of LMU in Munich is marked by a red dot and labeled with "MIM". Some more locations, e.g. the DLR in Oberpfaffenhofen, are marked for orientation.

which accounts for the difference between direct and apparent radiance due to the additional diffuse part and takes values $k \in [0,1]$. Using Eq. B5 the slant-path optical thickness $\tau_\mathrm{s}$ can be calculated by

$$\tau_\mathrm{s} = \ln\left(\frac{L_0}{L'}\right)/k = \tau_\mathrm{app}/k\,. \tag{B5}$$

As discussed in Reinhardt et al. (2014), for $\tau_\mathrm{s} < 3$ the correction factor $k$ is most sensitive to the detector FOV, the effective

particle radius and shape but is almost independent of $\tau_\mathrm{s}$ itself. For the retrieval the $k$-factors were calculated according to this procedure for the SSARA FOV of $1.2°$ (Toledano et al., 2009) assuming a COT of 1.5 as proposed by Reinhardt et al. (2014). The $k$-factors were calculated for all ice crystal habits, surface roughness values, and effective radii used in the LUT. The COT is then computed by dividing the apparent COT by $k$ (cf. Eq. B5) .

### B1.3 Surface albedo

The surface albedo is another parameter which affects the transmission measured at the ground, but its impact on the retrieval is significantly smaller compared to the aerosol type and optical thickness (cf. Fig. B1). With increasing surface albedo more radiation is reflected by the ground which is scattered back to the camera by the clouds. To estimate the surface albedo during the time of the measurements, the MODIS white-sky albedo product MCD43B3 (Strahler et al., 1999) was used. The MODIS white sky albedo product is available for seven wavelength bands centered at $469$, $555$, $645$, $858$, $1240$, $1640$ and $2130\,\mathrm{nm}$.

Figure B5 shows the MODIS white-sky albedo for the $555\,\mathrm{nm}$ wavelength band. The displayed geographic region was selected to cover the coordinates of the projected $22°$ halo between sunrise and sunset throughout the year. At a wavelength of $555\,\mathrm{nm}$ the albedo values range between about 0.015 and 0.12 with the lowest values for lakes (e.g. south of Starnberg) and forests (e.g. east of DLR). To obtain spectrally continuous data, the ASTER spectral library (Baldridge et al., 2009) is applied to interpolate the MODIS albedo data: a linear combination of the spectral albedo of deciduous and conifer trees, grass, shingle

and concrete is used to represent the MODIS white sky albedo. Figure B6 shows the MODIS white sky albedo measured at the seven wavelengths with black dots. The black line represents the linear combination of the single ASTER spectral albedos which provides the best match of the MODIS measurements. The single spectral albedos with the corresponding weighting coefficients are depicted in different colors. To obtain the albedo measured e.g. by $\mathrm{HaloCam_{RAW}}$, the fitted spectral albedo





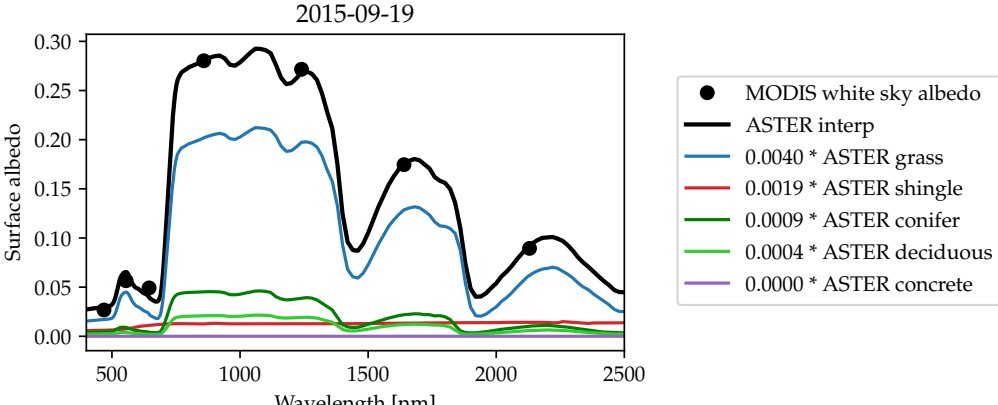

**Figure B6.** Spectral albedo data from the ASTER library provided with a resolution of 2 nm for grass (blue), shingle (red), conifer (dark green) and deciduous trees (green) as well as concrete (purple). A linear combination for the different ASTER albedo types is determined which represents best the averaged MODIS data from Fig. B5 by applying the least-squares method. The weighting factors for 19 September 2015 are provided in the legend of the figure. The resulting mixture of ASTER albedo data is then used to obtain an approximation of the MODIS albedo product for high spectral resolution, which is represented by the black solid line.

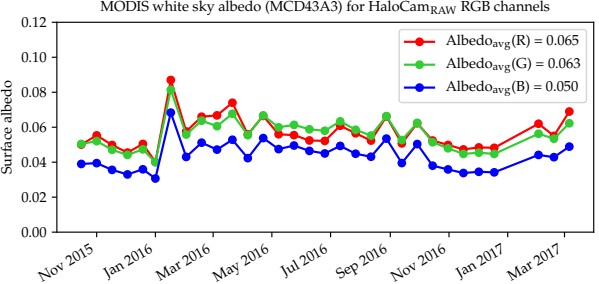

**Figure B7.** Surface albedo between October 2015 and March 2017 for the HaloCam$_{RAW}$ RGB channels. The data are obtained by weighting the spectral high-resolution parameterization of the MODIS albedo data (cf. Fig. B6 black curve) with the spectral response of the RGB channels (Forster et al., 2020). The surface albedo for the HaloCam$_{RAW}$ channels averaged over this period amounts to 0.065 (R), 0.063 (G), and 0.050 (B).

from the ASTER library (cf. black line in Fig. B6) is integrated over the spectral sensitivity of the respective camera channel. In the case of HaloCam$_{RAW}$, integrating the spectral albedo over the red, green, and blue channel yields the albedo values displayed in Fig. B7 with the respective line color. For this figure the MODIS white sky albedo values were evaluated between October 2015 and March 2017. Values larger than 0.1 were excluded since they are most likely due to snow cover. Averaging over the whole period yields mean albedo values for the red, green, and blue channel of 0.065, 0.063, and 0.050 respectively. The red and green channels show higher values than the blue channel since the surface South of Munich is dominated by green





grass and trees. Comparing the red and the green channel, a slight difference between winter and summer is noticeable which is very likely due to the vegetation period. During summer the deciduous trees increase the albedo in the part of the spectrum covered mostly by the green channel, whereas in winter the albedo measured by the green channel is slightly lower than the red channel.

*Code and data availability.* AERONET data for the station "Munich University" is available via https://aeronet.gsfc.nasa.gov/. Radiosonde observations for München-Oberschleißheim (station no. 10868) can be downloaded using http://weather.uwyo.edu/upperair/sounding.html. The MODIS white-sky albedo product MCD43A3 was retrieved from https://opendap.cr.usgs.gov/opendap/hyrax//MODV6_Cmp_B/MOTA/MCD43A3.006/. All websites were last accessed 19 January 2022. The look-up tables, image data, the retrieval algorithm, the institute's SSARA sun photometer and MIRA-35 cloud radar data used in this study will be provided upon request by the authors.

*Author contributions.* LF prepared the manuscript, developed the retrieval method and measurement strategy, pre-processed and calibrated the HaloCam dataset, compiled the DISORT look-up tables, and analyzed the retrieval results as part of her doctoral thesis. BM secured the funding for the HaloCam project and supervised the doctoral thesis. He provided valuable feedback on the retrieval method and data analysis, as well as on the manuscript.

*Competing interests.* The authors declare that no competing interests are present.

*Acknowledgements.* We thank Markus Rapp (DLR Oberpfaffenhofen) for co-funding the PhD thesis. We further acknowledge Matthias Wiegner for providing the AERONET sun photometer measurements, Meinhard Seefeldner, Markus Garhammer, and Anton Lex for their help with maintaining the sun photometers and HaloCam's sun-tracking mount. Florian Ewald and Tobias Zinner kindly provided the MIRA-35 cloud radar measurements. We thank Claudia Emde for implementing Yang et al.'s ice crystal optical properties in libRadtran and providing valuable feedback on the project.



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
