# Peer review of "Ice Crystal Characterization in Cirrus Clouds III: Retrieval of Ice Crystal Shape and Roughness from Observations of Halo Displays"

_Atmospheric Chemistry and Physics, 2022_

## Author Comment (AC1)

**Reply to comments by referee #1**

We thank the referee for taking the time to carefully review the manuscript. We appreciate the positive feedback and the valuable suggestions and comments, which are addressed below. The referee's comments are highlighted in blue.

1) Ice crystals in cirrus clouds are not monodisperse. Moreover, various ice crystal habits/shapes may coexist in a given cirrus cloud. In the revised manuscript, could the authors please elaborate on the impacts of the ice crystal size and shape distributions on the results?

We appreciate this suggestion and addressed it by adding the following statements to the manuscript starting on page 5, line 26:

"Note that the retrieved ice crystal effective radius, shape and SCF will depend on assumptions about the underlying particle distribution, since the bulk optical properties, as e.g. the extinction coefficient $\beta_{\text{ext}}$, are obtained by integrating the single scattering properties over Eq. 1."

"Ice crystals in cirrus clouds are known to follow multi-modal rather than monomodal size, shape, and surface roughness distributions. Therefore matching ice crystal properties could be retrieved for mixtures of arbitrary complexity. However, this study aims at finding the simplest ice crystal model with the minimum degrees of freedom that matches the observations within the measurement uncertainty. Inspired by Schmitt and Heymsfield (2014) and Liu et al. (2014), who separate the huge variety of ice crystal shapes into simple and complex crystals, we employ this two-habit approach for smooth and rough crystals to represent the "halo-producing" and "non-halo-producing" category of ice particles. "

2) In Eqs. (1) and (2), the extinction coefficient is involved, which is a bulk radiative quantity. But the particle size distribution is not specified.

We added information about the particle size distribution starting on page 5, line 17:

"The resulting ice crystal properties assumed here represent a single ice crystal shape, two levels of surface roughness, and follow a particle size distribution $n$ according to:

$$n(D) = D^{\nu} \exp(-\lambda D), \tag{1}$$

with maximum crystal dimension $D$ and $\nu = 1$ fixed. For a given effective radius $r_{\text{eff}}$, the optical properties provided for a range of maximum dimensions $D$ in YG13 were integrated over the size distribution. During integration, $\lambda$ was determined iteratively to match the computed with the prescribed effective radius. The smooth crystal fraction

$$\text{SCF} = \beta_{\text{ext,smooth}}/\beta_{\text{ext,total}}, \tag{2}$$

with $\beta_{\text{ext,total}} = \beta_{\text{ext,smooth}} + \beta_{\text{ext,rough}}$ ranges between $0 \leq \text{SCF} \leq 1$, resulting in a rough crystal fraction of

$$\text{RCF} = 1 - \text{SCF} = \beta_{\text{ext,rough}}/\beta_{\text{ext,total}}. \tag{3}$$

Note that the retrieved ice crystal effective radius, shape and SCF will depend on assumptions about the underlying particle distribution, since the bulk optical properties, as e.g. the extinction coefficient $\beta_{\text{ext}}$, are obtained by integrating the single scattering properties over Eq. 1."

3) As stated in the second paragraph on page 3, the authors' previous findings confirm that 25% of cirrus clouds produce 22-deg halos. However, spaceborne observations seem to suggest that ice crystals in cirrus clouds are roughened, for example, as demonstrated by one year of POLDER/PARASOL observations (see, Fig. 15 in Yang, P., S. Hioki, M. Saito, C.-P. Kuo, B. A. Baum, K.-N. Liou, 2018: A review of ice cloud optical property models for satellite remote sensing, Atmosphere 2018, 9, 499; doi:10.3390/atmos9120499). It will be valuable if this manuscript provides insight to coincide the finding based on spaceborne observations with that based on ground-based observations.

Our finding that a small fraction of smooth ice crystals mixed with severely roughened ice crystals is sufficient to produce a visible halo is not in contradiction to findings from spaceborne observations suggesting that ice crystals in cirrus clouds are roughened in general. We address this question in the discussion section of this manuscript starting on page 21 line 13, where we added the above mentioned publication:

"Our finding that columnar ice crystal shapes best represent the HaloCam observations further implies that a majority of rough ice crystals mixed with a smaller fraction of smooth crystals is sufficient to produce a visible 22° halo. Finding predominantly rough and complex ice crystals to best match the observations is in agreement with the results of several studies based on satellite retrievals. Using multi-angle reflectance measurements, Baran et al. (1998, 1999) and McFarlane and Marchand (2008) found polycrystals and complex crystals to better represent the observations than pristine single crystals. Studies based on multi-angular polarized reflectances from POLDER (Polarization and Directionality of Earth Reflectance) also report that featureless phase functions, which correspond to roughened or complex crystals, better represent the measurements than phase functions of a single ice crystal habit (Descloitres et al., 1998; Chepfer et al., 2001; Baran et al., 2001; Baran and Labonnote, 2006; Sun et al., 2006; Yang et al., 2018). Holz et al. (2016) and Wang et al. (2014) confirmed that rough and complex crystals better match the observations than smooth single crystals for optically thin clouds (COT < 3) using retrievals based on lidar observations and reflectances in the infrared spectrum."

Optional minor editorial revisions:

a) Lines 2-3 on page 1: change "making use of" to "using"

Changed.

b) Line 4 on page 1: Change " ... the retrieval of size and shape of randomly oriented crystals" to " ... the retrieval of the sizes and shapes of randomly oriented ice crystals"

Changed.

c) Line 12 on page 1: change "forward scattering part of the ice crystal optical properties" to "forward portion of the light scattered by ice crystals"

Changed.

d) Line 13 on page 3: change "...retrieve ice crystal shape and surface roughness" to "...retrieve ice crystal shape and the degree of surface roughness".

Changed.

e) Line 12 on page 4: change "To the authors' knowledge" to "To the best of the authors' knowledge".

Changed.

f) Line 32 on page 4: change "Look-up tables (LUT)" to "Look-up tables (LUTs)".

Changed.

**References**

Baran, A., Watts, P., and Francis, P.: Testing the coherence of cirrus microphysical and bulk properties retrieved from dual-viewing multi-spectral satellite radiance measurements, Journal of Geophysical Research, 104, 31 673–31 683, https://doi.org/10.1029/1999JD900842, 1999.

Baran, A., Francis, P., Havemann, S., and Yang, P.: A study of the absorption and extinction properties of hexagonal ice columns and plates in random and preferred orientation, using T-Matrix theory and aircraft observations of cirrus, J. Quant. Spectrosc. Radiat. Transfer, 70, 505–518, https://doi.org/10.1016/S0022-4073(01)00025-5, 2001.

Baran, A. J. and Labonnote, L. C.: On the reflection and polarisation properties of ice cloud, J. Quant. Spectrosc. Radiat. Transfer, 100, 41–54, https://doi.org/10.1016/j.jqsrt.2005.11.062, VIII Conference on Electromagnetic and Light Scattering by Nonspherical Particles, 2006.

Baran, A. J., Watts, P. D., and Foot, J. S.: Potential retrieval of dominating crystal habit and size using radiance data from a dual-view and multiwavelength instrument: A tropical cirrus anvil case, Journal of Geophysical Research, 103, 6075–6082, https://doi.org/10.1029/97JD03122, 1998.

Chepfer, H., Goloub, P., Riedi, J., De Haan, J. F., Hovenier, J., and Flamant, P.: Ice crystal shapes in cirrus clouds derived from POLDER/ADEOS-1, Journal of Geophysical Research, 106, 7955–7966, https://doi.org/10.1029/2000JD900285, 2001.

Descloitres, J., Buriez, J.-C., Parol, F., and Fouquart, Y.: POLDER observations of cloud bidirectional reflectances compared to a plane-parallel model using the International Satellite Cloud Climatology Project cloud phase functions, Journal of Geophysical Research, 103, 11 411–11 418, https://doi.org/10.1029/98JD00592, 1998.

Holz, R. E., Platnick, S., Meyer, K., Vaughan, M., Heidinger, A., Yang, P., Wind, G., Dutcher, S., Ackerman, S., Amarasinghe, N., Nagle, F., and Wang, C.: Resolving ice cloud optical thickness biases between CALIOP and MODIS using infrared retrievals, Atmos. Chem. Phys., 16, 5075–5090, https://doi.org/10.5194/acp-16-5075-2016, 2016.

Liu, C., Yang, P., Minnis, P., Loeb, N., Kato, S., Heymsfield, A., and Schmitt, C.: A two-habit model for the microphysical and optical properties of ice clouds, Atmos. Chem. Phys., 14, 13 719–13 737, https://doi.org/10.5194/acp-14-13719-2014, 2014.

McFarlane, S. A. and Marchand, R. T.: Analysis of ice crystal habits derived from MISR and MODIS observations over the ARM Southern Great Plains site, Journal of Geophysical Research: Atmospheres, 113, D07 209, https://doi.org/10.1029/2007JD009191, 2008.

Schmitt, C. G. and Heymsfield, A. J.: Observational quantification of the separation of simple and complex atmospheric ice particles, Geophysical Research Letters, 41, 1301–1307, https://doi.org/10.1002/2013GL058781, 2013GL058781, 2014.

Sun, W., Loeb, N. G., and Yang, P.: On the retrieval of ice cloud particle shapes from POLDER measurements, J. Quant. Spectrosc. Radiat. Transfer, 101, 435–447, https://doi.org/10.1016/j.jqsrt.2006.02.071, 2006.

Wang, C., Yang, P., Dessler, A., Baum, B. A., and Hu, Y.: Estimation of the cirrus cloud scattering phase function from satellite observations, J. Quant. Spectrosc. Radiat. Transfer, 138, 36–49, https://doi.org/10.1016/j.jqsrt.2014.02.001, 2014.

Yang, P., Hioki, S., Saito, M., Kuo, C.-P., Baum, B. A., and Liou, K.-N.: A Review of Ice Cloud Optical Property Models for Passive Satellite Remote Sensing, Atmosphere, 9, https://doi.org/10.3390/atmos9120499, 2018.

---

## Author Comment (AC2)

**Reply to comments by referee #2**

We appreciate the time and care the referee took to review the manuscript. The valuable suggestions and comments will be addressed below. The referee's comments are highlighted in blue.

**Major comments**:

Ice crystal habit, size and smooth crystal fraction (SCF) is inferred from the halo observations by matching the observations at different viewing angles to simulations. However, it is not shown how the halo properties change with habit, size and SCF. I do not believe this is described in the other two parts of the series. I suggest using the LUT to show how the observations vary systematically with habit, size and SCF. I suggest the investigations of sensitivities to size and SCF focus on the plate and

10 column aggregate, as these appear to match the data mostly while yielding quite different results on SCF as seen in Fig 4.

We appreciate this suggestion and added the following paragraph accompanied by an additional figure to illustrate how the halo properties, in this case represented by the 22° and 46° halo ratio, depend on ice crystal shape, effective radius and smooth crystal fraction (SCF):

[revised manuscript text omitted]

"

Specifically: on page 8 you say that cases with RMSE values below 2-sigma measurement uncertainty are "considered possible solutions", while "the LUT element with the minimum RMSE represents the best match." How are the possible solutions used? Isn't only the best match considered the retrieved parameter set? Are there instances where the lowest RMSE is not lower than the 2-sigma measurement uncertainty and thus the retrieval fails?

Thank you for pointing this out. We rephrased the sentences for clarification. Changes are highlighted in red in the paragraph above.

The RMSE values are not very useful without a reference point, so please give a representative value of this 2-sigma measurement uncertainty. Alternatively or additionally, RMSE values could be normalized by the 2-sigma measurement uncertainty. The range of RMSE values in Fig 3 is quite broad, so is the 2-sigma measurement uncertainty quite large? Maybe I am not interpreting the RMSE values in Fig 3 correctly and these are not only for the 'best match' cases? If so, please explain in the text.

The $2\sigma$ radiometric measurement uncertainty depends on the measured radiance values and will be different for every time step. Typical values are e.g. 10 $\mathrm{mW\,m^{-2}nm^{-1}sr^{-1}}$ for a measured radiance of 220 $\mathrm{mW\,m^{-2}nm^{-1}sr^{-1}}$, which corresponds to an absolute radiometric response of about 4.5% as provided in Tab. 3 (Forster et al., 2020). Since the retrieval is applied to each time step separately, the same $2\sigma$ threshold and range of $L_{\mathrm{meas,i}}$ will be the reference to determine the best matching look-up table element. Normalizing the RMSE, would only change the $x$-coordinate of Fig. 4, but would not change how the RMSE values compare between the different ice crystal shapes. Since an additional reference point or normalization would not change the results of the retrieval, we decided to leave it as is.

Then in Figure 3, the AOD differs between the results for different habits, but it was explained that the AOT is constrained by clearsky observations before or after the cirrus observations, so I do not understand why this varies with habit?

The AOT is constrained to an interval between $\pm 2\sigma$ of the average AOT for clearsky observations before and/or after the halo observation. Since it can only be measured during clea sky, we consider this the best estimate for the halo observations. Depending on the single scattering properties of the selected ice crystal habit, the best matching LUT element might correspond to a combination of COT and AOT with slightly different values, within their respective intervals. **Changes are highlighted in bold font in the paragraph above**.

On page 7, line 13, it is stated that "the LUT was also constrained to SSARA's COT measurements within a 2-sigma confidence interval averaged over a ±5 min time interval." I am not sure what is meant here. I assume the COT within the LUT that matches most closely to the mean SSARA's COT is used, but it is not clear how the 2-sigma confidence interval is used.

Only halo observations corresponding to COT values within this interval were considered. Amongst these observations, the best match was selected. The rest of the observations is excluded from the retrieval. Changes are highlighted in teal font in the paragraph above.

On page 13, the sensitivity study is described and it is stated that "the LUT was modified by multiplication with a slope". But what specifically was multiplied? And with a slope with respect to what? Please clarify. Also, does the modified LUT also lead to a different SCF?

We appreciate the feedback and modified the section in the manuscript to better explain the procedure. The retrieved SCF remains mostly unaffected by using the modified LUT for the retrieval, except for the urban aerosol case which slightly changed the retrieved SCF for plates, 5-element and 10-element plates as described in the changes below.

"To investigate the sensitivity of the retrieval to different aerosol types, we would ideally compute new LUTs. Since computing a new LUT would require several weeks computation time on MIM's high performance computing cluster for each new aerosol type and the radiance at each scattering angle would basically only differ by a multiplicative factor, we repeated the retrieval with a modified LUT to estimate the effect of these approximations. The LUT was modified by multiplication with a factor for each scattering angle, which is representative for the amount and the sign of the bias introduced by the approximations. The multiplicative factor for each scattering angle in the LUT, which we refer to as "slope" in the following, was computed by the ratio between two radiance distribution, simulated with DISORT: One "reference" radiance distribution using the continental average aerosol type and one "modified" radiance distribution for each of the aerosol types: continental clean, continental polluted, and urban. In addition, a slope was generated by computing the ratio between a "reference" radiance distribution accounting for HaloCam$_{RAW}$'s full spectral response (cf. solid red line in Fig. B3) and a "modified" radiance distribution based on HaloCam$_{RAW}$'s representative wavelength of 618 nm for the red channel (cf. dashed red line in Fig. B3). These slopes were computed for each of the eight ice crystal habits assuming a representative atmospheric setup: COT = 0.8, AOT = 0.1, and a SCF of 30% for columnar crystals, 60% for hollow column crystals, and 70% for plate-like crystals.Table 4 shows the results of the best matching habit for each day retrieved with the modified LUT. The best matching habit changed slightly for the different modifications of the LUT but only within the plate-like or column-like crystal groups. The ice crystal plates remain the overall best-matching habit in the considered scattering angle range. The retrieved SCF in Table 3 remained mostly unaffected by using the modified LUT for the retrieval. Only for the urban aerosol case, the retrieved SCF for plates changed from $(80 \pm 10)\%$ to $(70 \pm 10)\%$, for 10-element plates from $(70 \pm 10)\%$ to $(80 \pm 10)\%$ and for 5-element plates from $(70 \pm 20)\%$ to $(60 \pm 20)\%$. "

In section 3.2 the 46 degree halo is simulated, but it is unclear what the SCF and size is used here, i.e. in Figs 6 and 7.

Appyling the retrieval to halo observations averaged over a whole day is the best approach to answer the question whether a (faint) 46° halo was present during the observations but yields, as expected, different results compared to applying the retrieval to individual observations and then averaging the retrieved cirrus properties. For the first approach, varying AOT and COT values over the course of the observation period cannot be accounted for, however the retrieved SCF and effective radius are strongly correlated with these values. This retrieval approach can therefore only help to constrain the particle shape but might induce a bias in the retrieved SCF and effective radius. For this reason we did not mention these values here to avoid the misunderstanding that those were the final retrieval results. We added the following explanation to the manuscript and for the sake of completeness, the ice crystal properties used for the DISORT simulations in a footnote.

"As mentioned above, applying the retrieval to HaloCam observations, which were averaged over a whole day, yields only qualitative results that help us confirm which ice crystal shape best matches the region both of the 22° and 46° halo. Since it does not allow for the retrieved cirrus and aerosol optical thickness to follow their natural temporal fluctuation, the retrieved smooth crystal fraction and ice crystal radius might get biased[1]. We therefore repeated the quantitative retrieval as described in Section 3.1 for the individual HaloCam$_{RAW}$ images, but this time excluding all LUT elements with a 46° halo, corresponding to a halo ratio $> 1$ (cf. Fig 1)."
* * *
[1]For the sake of completeness, we provide here the ice crystal properties used for the DISORT simulations: Fig. 8b (SCF=80 %, $r_{eff}$=10$\mu$m), Fig. 9b (SCF=20 %, $r_{eff}$=10$\mu$m), Fig. 9e (SCF=20 %, $r_{eff}$=10$\mu$m), Fig. 9h (SCF=20 %, $r_{eff}$=10$\mu$m), Fig. 9k (SCF=80 %, $r_{eff}$=5$\mu$m), Fig. 9n (SCF=20 %, $r_{eff}$=10$\mu$m).

The LUT elements "with a 46 degree halo" are excluded, but how is this quantified?

LUT elements for which the 46° halo ratio is > 1 are excluded. We added this statement to the manuscript.

"The retrieval was repeated for the individual HaloCam$_{\mathrm{RAW}}$ images excluding all LUT elements with a 46° halo, corresponding to a halo ratio > 1 (cf. Fig. 1)."

**Minor comments**:

Page 1, Line 21: add "be" between "to" and "more"

Done.

10  Page 2, line 4: add "to" after "helps"

Done.

Page 2, line 21: The papers van Diedenhoven et al. (2012, 2020) are cited referring to RSP data, but both papers use POLDER data. The van Diedenhoven et al. (2013) that is also in the reference list is using RSP data and could be cited here as well. The other two papers could be cited in the previous sentence, although "more recently" does not apply then anymore.

15      We corrected this section:

"Multi-angular polarized reflectances from the Polarization and Directionality of Earth Reflectance (POLDER) have been used to infer information about ice crystal shape (e.g. Descloitres et al., 1998; Chepfer et al., 2001; Baran and Labonnote, 2006; Sun et al., 2006). More recently, POLDER observations have been used to retrieve ice crystal aspect ratio and distortion levels: van Diedenhoven et al. (2012, 2020); van Diedenhoven (2021) found that crystal distortion and aspect ratio increase with cloud top height, leading to decreasing asymmetry parameters. These studies mainly focus on tops of optically thick ice clouds."

Page 4, line 3-4: : ice crystal orientation also has significant effects on the global radiative budget (Noel and Sassen, 2005)". This is an overstatement and not supported by the cioted paper. Other studies, such as Breon and Dubrulle (https://doi.org/10.1175/JAS-3309.1) and Zhou et al. (https://doi.org/10.1175/JAMC- D-11-0265.1) have concluded that the percentage of oriented plates in 25  the clouds is verry low and that "These low fractions imply that the impact of oriented plates on the cloud albedo is insignificant." Please correct the statement in the paper.

Thank you for the pointer. We removed this part of the statement:

"While ice crystal orientation also has significant effects on the remote sensing of ice cloud properties, this study focuses on randomly oriented ice crystals for a start and leaves investigation of oriented crystals for a future study."

30  Page 5, line 9 and throughout the paper: The 8-element aggregate of columns is named "8-element columns" here, but this is confusing in my view. Please refer to this as an aggregate throughout the paper. Also correct the naming of the plate aggregates accordingly.

We corrected the naming in the abstract and conclusions as well as for the first time describing these crystal shapes, but refer to the short version throughout the main part of the manuscript for the sake of brevity. We added a note in the text:

35      "Optical properties based on Yang et al. (2013) (referred to as YG13 in the following) were used for eight different habits: solid columns, hollow columns, plates, 8-element aggregate of columns, 5-element aggregate of plates, 10-element aggregate of plates, solid bullet rosettes, and hollow bullet rosettes, all of which are based on hexagonal crystal symmetry. Droxtals were not considered for the retrieval since they do not produce a 22° halo (Yang et al., 2013). For the sake of brevity, we will refer to the aggregates of columns and plates as 8-element columns, 5-40      element plates, and 10-element plates."

Page 5, line 26: A bias in what? Radiance, retrieved SCF? Please specify.

We clarified this statement:

" Figure B3 in Appendix B shows that this causes a bias in the 22° halo radiances of 1.5% for the blue, 2.0% for the green, and 1.2% for the red channel."

5 Page 10, line 3: The grammar of this sentence does not seem to be correct.

Corrected.

"For solid columns (Fig. 5a) the SCF peaks below 50% and HaloCam$_{RAW}$'s 22° halo observations are represented best by a mean SCF of 35.9% and RCF of 64.1% (cf. Eq. 3)."

Page 10, line 35: Note that the base temperature of -10C is out of the range of your definition of cirrus.

10    Thank you for pointing this out. We corrected that statement:

"In both the 2017 and the present study we refer to cirrus clouds as non-precipitating ice clouds. In the 2017 study, we even constrained the observations to cloud base temperatures of $-20\,°$C or colder."

We kept the order of figures 6 and 7 (now 8 and 9) but adjusted the order of reference in the text. This section now reads:

15    "For analyzing this scattering angle region, we therefore use HaloCam$_{RAW}$ observations averaged over each day and make use of the presence or absence of the 46° halo in a qualitative way to further constrain the retrieved ice crystal properties from Section 3.1. We focused this analysis on six of the eight days, for which the number of halo samples was high and the horizontal extent of the cirrus cloud was large enough to yield homogeneous conditions across both the 22° and 46° halo regions in the averaged image. If ice crystals in the cirrus cloud were able to
20    form a 46° halo, we would expect to see it in the averaged image. Figure 8 displays the averaged HaloCam$_{RAW}$ measurements for 22 September 2015 (a) in comparison with DISORT simulations (b) using ice crystal plates, which were found to best match the observations in the region of the 22° halo (cf. Table 3). "